# EFDIFF: FREQUENCY-INFORMED DIFFUSION FOR EXTREME-VALUE TIME SERIES GENERATION

## ABSTRACT

Time-series generation, which aims to produce realistic synthetic sequences that preserve temporal dynamics, is essential for data augmentation and practical applications. However, existing methods often fail to capture extreme-value distributions, which are crucial in domains such as finance, climate, and energy. This limitation mainly stems from overall-fit objectives and smoothing procedures that distort extreme-event structures. To address these challenges, we propose EFDiff, a frequency-informed extreme-aware time-series generation framework. Unlike conventional approaches that focus on long-tail preservation in the time domain, EFDiff adopts a frequency-domain perspective by integrating a frequency-based disentanglement strategy into diffusion models. The key innovation lies in an Extreme Component, which consists of two key modules: (i) Extreme-Frequency Extraction (EFX), which constructs a global extreme-frequency dictionary that characterizes potential extreme patterns via event-driven local analysis and multimetric integration based on the proposed concept of extreme-contributing frequencies; and (ii) Extreme-Frequency Generation Enhancement (EFGEN), which includes a novel Transformer-based Soft Frequency Selection Network to identify relevant frequencies and effectively model extreme patterns during the denoising process. Extensive experiments on five real-world datasets across six evaluation metrics demonstrate that EFDiff consistently achieves strong overall generation quality and substantially improves the fidelity of extreme-value generation.

## 1 INTRODUCTION

Time series constitute one of the most important data types in real-world applications, spanning domains such as climate science, finance, healthcare, energy, and manufacturing Nikitin et al. (2024). Recently, time-series generation has attracted increasing attention for its strong effectiveness in improving data quality, enriching diversity, and preserving privacy Yoon et al. (2019); Jarrett et al. (2021); Alaa et al. (2021a); Naiman et al. (2024); Chen et al. (2024); Yuan & Qiao (2024). In time-series data, an important aspect is the occurrence of extreme events, defined as observations located in the tails of a probability distribution and representing rare yet impactful deviations from typical behavior Gu et al. (2025). However, most existing generative models focus primarily on overall generation quality, e.g., emphasizing global distributional fidelity, and their architectures inherently suppress outlier patterns during training and denoising, thereby smoothing out extreme events Yang et al. (2023); Falck et al. (2025), which may cause significant information loss in the synthetic data.

Although some recent studies try to strengthen generative models' ability to capture rare or extreme events within heavy-tailed distributions Bhatia et al. (2021); Huster et al. (2021); Allouche et al. (2022a); Pandey et al. (2025), these approaches remain limited, as they mainly increase the likelihood of extreme events by emphasizing heavy-tailed distributions, without incorporating the underlying contexts and patterns of such events Gu et al. (2025). Since extreme-value structures arise from the phase alignment of critical high-frequency components, it is insufficient to focus only on dominant high-amplitude structures but necessary to consider more complex and unstable high-frequency components that collectively contribute to the formation of extremes. Therefore, to advance existing research, in this paper, we investigate the time-series generation from a frequency-domain perspective, which offers two key opportunities for extreme-value modeling. First, frequency-domain analysis supports multi-scale decomposition representations as low frequencies capture long-term trends, mid frequencies reflect periodic structures, and high frequencies highlight extreme events.

This separation also enables clearer disentanglement of frequencies in time-series data, which is crucial for extreme-value generation. Second, frequency-domain analysis incorporating phase information that is invisible in the time domain is essential for understanding the formation of extreme values. However, two critical challenges remain: (1) it is difficult to identify the specific frequencies that contribute most to extreme values, and (2) it remains unclear how to effectively leverage these frequencies in generative models to better capture extreme values for generation.

To address these challenges, we propose EFDiff, a frequency-informed diffusion framework for extreme-aware time-series generation. (i) In EFDiff, each sequence is first decomposed into three components during sampling: trend, seasonality, and extreme component, corresponding to the low-, mid-, and high-frequency parts. (ii) We then propose Extreme-Frequency Extraction (EFX), which outputs a global extreme-frequency dictionary comprising diverse frequency combinations that characterize potential extreme patterns based on event-driven local analysis and multi-metric integration. Combinatorial reconstruction is also employed to capture phase alignment across different frequencies. (iii) We further design Extreme-Frequency Generation Enhancement (EFGEN) to recover extreme events during the denoising process. The core of EFGEN is a Transformer-based Soft Frequency Selection Network, which automatically selects the most suitable extreme patterns from the extreme-frequency dictionary while determining their amplitudes and phases simultaneously.

The contributions of this paper are summarized as follows:

- We propose a frequency-domain perspective for extreme-value time-series generation, leveraging the phase alignment of high-frequency components to effectively model complex extreme patterns. Through data analysis, we further uncover key insights into the relationship between extremes and frequencies, which guide the subsequent design of our framework.

- Revised version: We introduce EFDiff, a frequency-informed diffusion framework for extreme-aware time-series generation. Its central novelty lies in the Extreme Component with two modules: Extreme-Frequency Extraction (EFX), which captures overall extreme patterns, and Extreme-Frequency Generation Enhancement (EFGEN), that enhance extreme structure during denoising.

- Extensive experiments on five real-world datasets and six metrics validate the effectiveness of our method. The results demonstrate that EFDiff consistently delivers strong overall generation quality while substantially enhancing the fidelity of extreme-value generation. For example, EFDiff yields a 55.7% KL Divergence improvement on the Stocks dataset over the best baseline. All code and data of this paper can be found in the anonymous **GitHub repository**.

## 2 DATA-DRIVEN ANALYSIS

Based on analysis of both synthetic and real-world data, we reveal the data-driven insight into the relationship between extremes and frequencies, which guides our subsequent framework design.

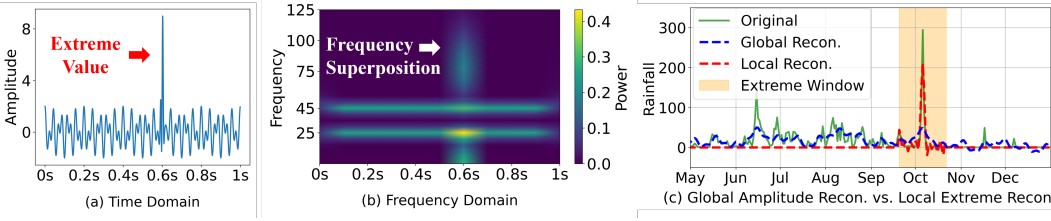

Figure 1: (a)(b) Visualization for extreme value in the time domain and the frequency domain, respectively. (c) Global amplitude-based reconstruction vs. local contribution-based reconstruction of extreme values.

**Insight: In the frequency domain, extreme values often arise from the superposition of multiple high-frequency components under specific phase alignments.** While low- and mid-frequency components usually dominate with larger amplitudes and shape trends and seasonalities, high-frequency interactions are critical for extreme values. In Figure 1, we illustrate this phenomenon with an example in both time and frequency domains. Figure 1(a) shows a signal formed by sine waves at 25 Hz and 45 Hz with a sharp disturbance at 0.6 s to emulate an extreme. Figure 1(b) presents the corresponding frequency-domain representation obtained using the Short-Time Fourier

Transform (STFT), where horizontal bands reflect persistent 25 Hz and 45 Hz energy, and the vertical column near 0.6 s reveals the coexistence of multiple high frequencies. This observation indicates that, beyond capturing the dominant time-series patterns (25 Hz and 45 Hz), effectively reproducing extreme events requires incorporating the superposition of multiple high-frequency components. Figure 1(c) compares data with various reconstruction strategies: the green line is the original sequence, the blue line shows the amplitude-based reconstruction using the top-16 frequencies by amplitude, and the red line is the contribution-based reconstruction using the top-16 frequencies most relevant to the extreme. We found that while the amplitude-based approach follows the original sequence, it fails to capture extremes in the highlighted yellow region, whereas the contribution-based approach provides a more faithful reconstruction for the extreme event.

From the data-driven insight, focusing only on high-amplitude frequencies that capture dominant patterns is insufficient. This raises two key research questions: (1) How can we assess the contributions of different frequencies to extreme values? (2) How can we design a model that leverages these extreme-contributing frequencies for more effective generation?

## 3 PRELIMINARY

### 3.1 EXTREME VALUES AND EVENTS

In this section, we define extreme values and explain how they are clustered into extreme events. There are two typical approaches to deciding extreme values in Extreme Value Theory (EVT) Coles et al. (2001). The block maxima (BM) method partitions a time series into blocks and selects the maximum from each, but it may miss cases where extremes are densely concentrated. The peaks-over-threshold (POT) method models observations above a high threshold, but the fixed threshold often lacks flexibility. To address their limitations, we propose Peaks Over Dynamic Threshold within blocks (PODT), a hybrid strategy that combines the BM principle with adaptive thresholding to capture multiple extremes per block. The dynamic threshold $\tau_w^{\text{ext}}$ for block $w$ is defined as:

$$\tau_w^{\text{ext}} = \max\left(\alpha|\mu_w|,\ k_\sigma\sigma_w,\ k_{mad}\,\text{MAD}_w,\ Q_{1-p}(d_t)\right), \tag{1}$$

where $\mu_w$ is the block mean ($\alpha|\mu_w|$ as the mean-ratio threshold); $\sigma_w$ is the block standard deviation ($k_\sigma\sigma_w$ as the variance-based threshold); $\text{MAD}_w$ is the median absolute deviation ($k_{mad}\,\text{MAD}_w$ as the MAD-based threshold); and $Q_{1-p}(d_t)$ is the $(1-p)$ quantile of deviations $d_t = |x_t - \mu_w|$ (top-$p$ quantile threshold). The final threshold $\tau_w^{\text{ext}}$ is set as the maximum of the four candidates, adapting to intra-block variation and ensuring that only sufficiently deviated points are identified as extreme values. We then define **extreme values** $x_t^{\text{ext}}$ as those exceeding $\tau_w^{\text{ext}}$.

To avoid interference from nearby extreme values, we further define an **extreme event** as a sequence of consecutive extreme values within a short time window (e.g., multiple heavy rainfall instances in close succession). Hence, adjacent extreme values with the same direction and close temporal proximity are clustered into a single event. Within each event, the extreme value with the largest amplitude is chosen as the representative, so the **extreme event** $E^{\text{ext}}$ is described as follows:

$$E^{\text{ext}} = \left(t_{\text{rep}}^{\text{ext}},\ A_{\text{rep}}^{\text{ext}},\ W^{\text{ext}},\ \mathcal{I}^{\text{ext}}\right), \quad \mathcal{I}^{\text{ext}} = \left[\,t_{\min}^{\text{ext}} - r^{\text{ext}},\ t_{\max}^{\text{ext}} + r^{\text{ext}}\,\right], \tag{2}$$

where $E^{\text{ext}}$ denotes an extreme event represented as a quadruplet. $t_{\text{rep}}^{\text{ext}}$ is the representative time index of the event, and $A_{\text{rep}}^{\text{ext}}$ is the amplitude of its representative extreme value. $W^{\text{ext}}$ is the overall weight of the event, defined as the aggregated amplitude of all extreme values it contains. $\mathcal{I}^{\text{ext}}$ specifies the temporal span of the event, bounded by $t_{\min}^{\text{ext}}$ (the earliest extreme value) and $t_{\max}^{\text{ext}}$ (the latest extreme value), each extended by a margin $r^{\text{ext}}$.

### 3.2 FREQUENCY-INFORMED PRESPECTIVE

We define $X_{1:T} = (x_1, \ldots, x_T) \in \mathbb{R}^T$ as a single time series of length $T$. Let $\mathcal{D} = \{X_{1:T}^i\}_{i=1}^N$ denote a dataset of $N$ sequences. Our goal is to develop a diffusion-based generative model that captures the patterns underlying extreme events and produces synthetic sequences that not only follow the overall distribution of $\mathcal{D}$ but also preserve the statistical characteristics of extremes.

In EFDiff, we study extreme event generation from a frequency-domain perspective. Specifically, we describe an **extreme event** from the frequency perspective, in which the representative extreme

value can be expressed as the constructive superposition of specific frequency components with nearly aligned phases. This relationship can be formally expressed as:

$$x_{t_{rep}}^{\text{ext}} = \sum_{f_m^{\text{ext}} \in \mathcal{F}^{\text{ext}}} A_m \cos\left(2\pi f_m^{\text{ext}} t_{rep} + \phi_m\right), \qquad \text{s.t.} \max_{f_m^{\text{ext}}, f_n^{\text{ext}} \in \mathcal{F}^{\text{ext}}} \left|\phi_m - \phi_n\right| \leq \varepsilon. \quad (3)$$

Here, we define $f_m^{\text{ext}}$ as an **extreme-contributing frequency**, i.e., a frequency whose contribution to extreme values exceeds a given threshold. Let $\mathcal{F}^{\text{ext}}$ denote the set of such frequencies. A key constraint is that the phases $\phi_m$ of all frequencies in $\mathcal{F}^{\text{ext}}$ remain nearly aligned, enabling their contributions to add coherently and amplify the signal to produce an extreme value.

## 4 PROPOSED FRAMEWORK: EFDIFF

In this section, we present the proposed framework EFDiff. First, we outline the overall perspective of the frequency-based disentanglement strategy and its application to constructing the original sample prediction network. Second, we introduce the key design of the Extreme Component through three subsections: Section 4.1 describes EFX, which extracts extreme-contributing frequencies; Section 4.2 details EFGEN, which leverages these frequencies to better capture extreme patterns during the denoising process; and Section 4.3 presents the overall loss function and explains how it guides the optimization of EFDiff. An overall framework of EFDiff is illustrated in Figure 2.

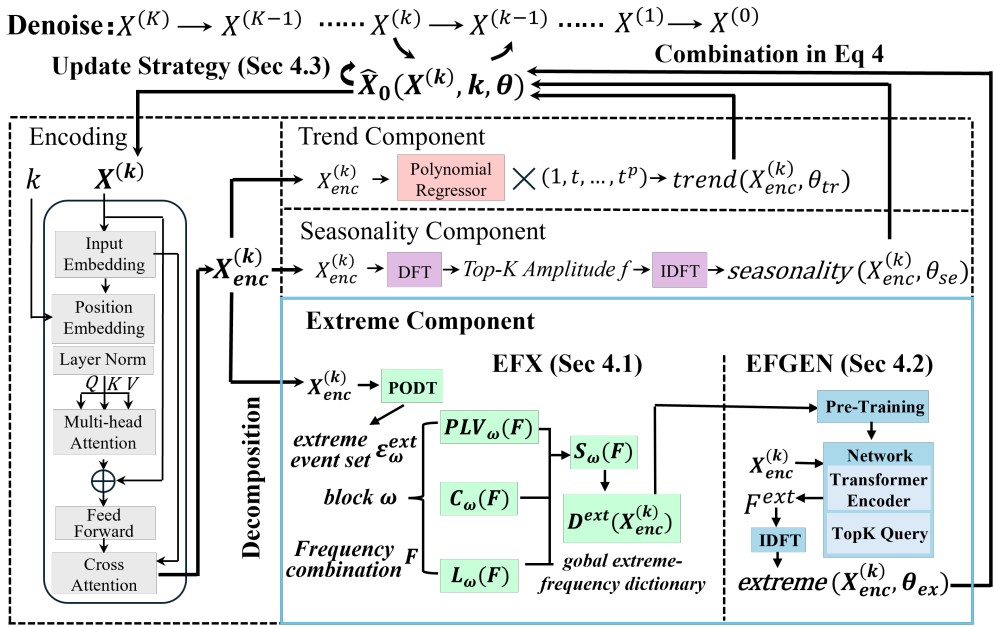

Figure 2: The Framework of EFDiff

In this work, we advance existing seasonal–trend decomposition in time-series generation Cleveland et al. (1990); Liu et al. (2023) by proposing an extreme frequency-aware disentanglement strategy. Our intuition is that low frequencies typically capture long-term trends, mid frequencies reflect periodic patterns, and high frequencies encode extreme information and noise. Based on this, the decomposition is constructed as:

$$\hat{X}_0(X^{(k)}, k, \theta) = trend\left(X_{enc}^{(k)}, \theta_{tr}\right) + seasonality\left(X_{enc}^{(k)}, \theta_{se}\right) + extreme\left(X_{enc}^{(k)}, \theta_{ex}\right) + \varepsilon, \quad (4)$$

where sample prediction function $\hat{X}_0(X^{(k)}, k, \theta)$ directly estimates the clean sample, $\varepsilon$ denotes the residual error term, and $X_{enc}^{(k)}$ represents the encoded version of $X^{(k)}$. In EFDiff, our primary focus is on constructing the **Extreme Component**, while the trend and seasonality parts follow the settings of the state-of-the-art Diffusion-TS Yuan & Qiao (2024). Specifically, the trend is modeled with a polynomial regressor (as shown in Eq. 5), and the seasonality is obtained by applying the discrete

Fourier transform (DFT) to decompose $X_{enc}^{(k)}$ (as shown in Eq. 6).

$$trend(X_{enc}^{(k)}, \theta_{tr}) = \phi(t)^\top \left(W_{tr}\, h(X_{enc}^{(k)}; a_{tr}) + b_{tr}\right), \tag{5}$$

$$seasonality\left(X_{enc}^{(k)}, \theta_{se}\right) = \mathcal{IDFT}(\mathcal{M}_K(\mathcal{DFT}(X_{enc}^{(k)}), \theta_{se})), \tag{6}$$

where $X_{enc}^{(k)}$ is the encoded noisy sequence at step $k$, $\phi(t) = [1, t, t^2, \ldots, t^p]^\top$ is a polynomial basis, $h(\cdot; a_{tr})$ is a neural feature extractor with parameters $a_{tr}$, and $(W_{tr}, a_{tr}, b_{tr})$ are learnable coefficients forming $\theta_{tr}$. $\mathcal{DFT}$ and $\mathcal{IDFT}$ denote the discrete Fourier transform and its inverse, $\mathcal{M}K(\cdot, \theta_{se})$ selects the Top-$K$ frequencies, and $\theta_{se}$ is a Transformer network refining their contributions. More details can be found in Appendix E.

### 4.1 EXTREME-FREQUENCY EXTRACTION

In this part, our focus is to construct the **Extreme Component** $extreme(X_{enc}^{(k)}, \theta_{ex})$ in Eq. 4. Our primary objective is to address the key question: how to identify the extreme-contributing frequencies? To this end, we propose Extreme-Frequency Extraction (EFX), a novel algorithm that takes original time-series samples as input and generates a global extreme-frequency dictionary, which consists of diverse frequency combinations that capture potential extreme patterns.

We consider the original time-series collection $\mathcal{D}$ as the training set, where $\mathcal{D} = \{X_{1:T}^i\}_{i=1}^N$. For each sample $X_{1:T} = (x_1, \ldots, x_T)$, we first pass $X$ through an encoding process to obtain its representation $X_{enc}$. Next, $X_{enc}$ is partitioned into multiple blocks. Within each block $w$, we apply our proposed method PODT in Section 3.1 to identify extreme values. These values are then clustered into extreme events, yielding the set $\mathcal{E}_w^{ext} = \{E_1^{ext}, E_2^{ext}, \ldots, E_M^{ext}\}$. Each extreme event is represented as $E^{ext} = \left(t_{rep}^{ext}, A_{rep}^{ext}, W^{ext}, \mathcal{I}^{ext}\right)$, denoting its representative time index, representative amplitude, overall weight, and temporal span.

In EFX, we define three extreme-contribution metrics to evaluate the frequencies in each block $w$ using the extreme event set $\mathcal{E}_w^{ext}$. To account for the phase alignment in Eq. 3, we adopt a combinatorial reconstruction strategy Volk et al. (2018), which evaluates frequency combinations rather than individual frequencies, as combinations naturally share a unified phase. Therefore, in EFDiff, extreme-contributing frequencies are represented in the form of frequency combinations.

First, we propose the concept of **Phase-Locking Value (PLV)**, which measures the degree to which a group of frequency combinations maintain stable phase alignment across the extreme events in the set $\mathcal{E}_w^{ext}$. In other words, we seek frequency combinations that consistently contribute in the same direction across most extreme events. A high PLV indicates concentrated phase angles, producing a stable constructive effect on extremes, whereas a low PLV reflects dispersed phases, where constructive and destructive contributions offset each other, leading to no stable overall effect on extreme events. Formally, for a frequency combination $\mathcal{F}$ within block $w$, the PLV is defined as:

$$\mathrm{PLV}_w(\mathcal{F}) = \left| \frac{1}{|\mathcal{E}_w^{ext}|} \sum_{E^{ext} \in \mathcal{E}_w^{ext}} \exp\left(i\, \theta_{\mathcal{F}}(t_{rep}^{ext})\right) \right|, \tag{7}$$

where $\mathcal{E}_w^{ext}$ is the extreme event set, and $\theta_{\mathcal{F}}(t_{rep}^{ext})$ denotes the instantaneous phase of the frequency combination $\mathcal{F}$ at the representative time index $t_{rep}^{ext}$ of the extreme event $E^{ext}$. This formulation is theoretically grounded in circular statistics, in which each phase is represented as a unit vector and the mean resultant length captures their overall concentration Fisher (1995).

Second, we introduce the **Instantaneous In-phase Contribution** ($C_w$), which measures how strongly a frequency combination contributes to the extreme events in $\mathcal{E}_w^{ext}$ at their representative times. Unlike PLV, which emphasizes phase coherence across events, $C_w$ captures instantaneous amplitude alignment at each event. In other words, PLV reflects phase synchrony, whereas $C_w$ quantifies amplitude-based contributions. The formal definition of $C_w$ is given as follows:

$$C_w(\mathcal{F}) = \left(\sum_{E^{ext} \in \mathcal{E}_w^{ext}} W^{ext} A_{rep}^{ext} [\, s^*(E^{ext}) \cos(\theta_{\mathcal{F}}(t_{rep}^{ext}))\,]_+\right) \Big/ \left(\sum_{E^{ext} \in \mathcal{E}_w^{ext}} W^{ext}\right) \tag{8}$$

where $s^*(E^{ext}) \in \{+1, -1\}$ indicates the polarity of each extreme event (positive or negative peak), and $[x]_+ = \max(0, x)$ retains only constructive contributions. The definition of $C_k$ is motivated by time–frequency energy analysis Boashash (2015), where a frequency's effective contribution is captured by the product of its instantaneous amplitude and in-phase component.

Third, we define the **Event-to-Background Contrast** ($L_w$), which measures how strongly a frequency combination contributes to extreme events relative to the surrounding background. While the first two measurements capture phase and amplitude contributions in extreme events, $L_w$ contrasts contributions between extreme-event windows and background windows, thereby assessing whether a frequency combination plays a discriminative role in extremes. We then define $L_w$ as:

$$\mu_{\text{evt}}(\mathcal{F}) = \frac{1}{|\mathcal{L}^{\text{ext}}|} \sum_{t \in \mathcal{L}^{\text{ext}}} |x_{\mathcal{F}}(t)|, \quad \mathcal{L}^{\text{ext}} = \bigcup_{E^{\text{ext}} \in \mathcal{E}_w^{\text{ext}}} \mathcal{I}^{\text{ext}}(E^{\text{ext}}), \tag{9}$$

$$\mu_{\text{bg}}(\mathcal{F}) = \frac{1}{|\mathcal{B}_w|} \sum_{t \in \mathcal{B}_w} |x_{\mathcal{F}}(t)|, \quad \mathcal{B}_w = w \setminus \mathcal{L}^{\text{ext}}. \tag{10}$$

$$L_w(\mathcal{F}) = \frac{1}{2}\left( \frac{\mu_{\text{evt}}(\mathcal{F}) - \mu_{\text{bg}}(\mathcal{F})}{\mu_{\text{evt}}(\mathcal{F}) + \mu_{\text{bg}}(\mathcal{F}) + \varepsilon} + 1 \right), \quad \varepsilon > 0 \tag{11}$$

where $\mu_{\text{evt}}(\mathcal{F})$ and $\mu_{\text{bg}}(\mathcal{F})$ denote the average magnitudes of the frequency combination $\mathcal{F}$ over extreme-event and background windows, respectively. $|x_{\mathcal{F}}(t)|$ denotes the instantaneous magnitude of the narrowband reconstruction of the frequency combination $\mathcal{F}$ at time $t$, $\mathcal{L}^{\text{ext}}$ is the union of the temporal spans of all extreme events in block $w$, and $\mathcal{B}_w$ is the complementary background set obtained by removing $\mathcal{L}^{\text{ext}}$ from block $w$. The small constant $\varepsilon > 0$ ensures numerical stability.

Based on the three contribution metrics, we define a joint scoring function $S_w(\mathcal{F})$ with two learnable parameters to evaluate which frequency combinations in block $w$ should be regarded as extreme-contributing frequencies (as shown in Eq. 12). The parameters $\alpha$ and $\beta$ are embedded as learnable coefficients and are adaptively optimized via gradient-based learning. For each block $w$ of the encoded sample $X_{\text{enc}}$, we rank all frequency combinations by their joint scores and select the top-$K$. These selected combinations are subsequently aggregated across blocks to construct the global extreme-frequency dictionary (as shown in Eq. 13).

$$S_w(\mathcal{F}) = \alpha \, \text{PLV}_w(\mathcal{F}) + \beta \, C_w(\mathcal{F}) + (1 - \alpha - \beta) \, L_w(\mathcal{F}), \qquad \alpha, \beta \geq 0. \tag{12}$$

$$\mathcal{D}^{\text{ext}}(X_{enc}) = \left\{ w \mapsto \text{Top-}K\{\mathcal{F} \mid S_w(\mathcal{F})\} \mid w \subset X_{enc} \right\}. \tag{13}$$

Finally, in EFX we obtain the global extreme-frequency dictionary $D^{ext}(X_{enc})$, which reveals the relationship between the sample $X_{enc}$ and its potential extreme-contributing frequencies. Next, we introduce EFGEN, which reveals how these extreme-contributing frequencies can be leveraged during the diffusion sampling process to better reconstruct extreme events.

## 4.2 EXTREME-FREQUENCY GENERATION ENHANCEMENT

After obtaining the global extreme-frequency dictionary $D^{\text{ext}}(X_{\text{enc}})$, we propose Extreme-Frequency Generation Enhancement (EFGEN), a novel algorithm designed to extract extreme-related information from each encoded input $X_{\text{enc}}^{(k)}$. To achieve this, we develop a Transformer-based Soft Frequency Selection Network, which is first pre-trained on the dictionary generated by EFX and then fine-tuned during the diffusion denoising process. Here, soft selection refers to assigning adaptive weights to candidate frequency combinations rather than making discrete hard choices. In this way, pre-training enables the network to acquire an initial frequency selection strategy from real data. During fine-tuning, the network selects from a predefined frequency vocabulary those combinations most likely associated with extreme events and transforms them into the extreme component $extreme(X^{(k)}enc, \theta ex)$. As shown in Eq. 4, this component contributes to the construction of the sample prediction function $\hat{X}_0(X^{(k)}, k, \theta)$, which is subsequently employed in the diffusion denoising process to facilitate parameter updates.

Specifically, given an encoded input sequence $X_{\text{enc}}^{(k)}$ of length $L$, each observation $x_j$ in $X_{\text{enc}}^{(k)}$ is projected into a $d$-dimensional embedding with positional encoding PE:

$$z_j = \text{Linear}(x_j) + \text{PE}(j), \quad j = 1, \dots, L. \tag{14}$$

The sequence $Z = (z_1, \dots, z_L)$ is then processed by a multi-layer Transformer encoder to capture contextual representations. To extract frequency-related information, we introduce $K$ learnable query tokens $\{q_j\}_{j=1}^K$, which attend to the encoded sequence via cross-attention. Here, $K$ denotes the number of frequency combinations to be selected. The frequency vocabulary is defined as:

$$\mathcal{FV} = \{\mathcal{F}_1, \mathcal{F}_2, \dots, \mathcal{F}_M\}, \quad \mathcal{F}_m \subseteq \mathcal{B}, \tag{15}$$

where each entry $\mathcal{F}_m$ denotes a candidate frequency combination retrieved as the value linked to keys in the global extreme-frequency dictionary whose representations are most aligned with $X_{\text{enc}}$. Each query token produces a probability distribution over this vocabulary with a small adjustment parameter. Aggregating the contributions of all query tokens yields a score $s_m$ for each candidate $\mathcal{F}_m$. Finally, the top-$K$ extreme-contributing frequency combinations are selected as:

$$\mathcal{F}^{\text{ext}} = \text{Top-}K\big(\{(s_m, \Delta_m)\}_{m=1}^{M}\big), \tag{16}$$

where $s_m$ denotes the estimated probability that vocabulary entry $S_m$ contributes to extreme events, and $\Delta_m$ is its small offset.

For the selected set of extreme-contributing frequencies $\mathcal{F}^{\text{ext}}$, we construct the Extreme Component using the inverse discrete Fourier transform (IDFT):

$$\text{extreme}(X_{\text{enc}}^{(k)}, \theta_{ex}) = \sum_{f^{\text{ext}} \in \mathcal{F}^{\text{ext}}} A^{(f^{\text{ext}})} \exp\Big(i\Big(2\pi \frac{f^{\text{ext}}}{T} t + \Phi^{(f^{\text{ext}})}\Big)\Big), \quad t = 0, \dots, T-1, \tag{17}$$

where $A^{(f^{\text{ext}})} = \big|\hat{x}[f^{\text{ext}}]\big|$ and $\Phi^{(f^{\text{ext}})} = \arg\big(\hat{x}[f^{\text{ext}}]\big)$ denote the amplitude and phase of the selected Fourier component, respectively. Here, $\theta_{ex}$ is the set of learnable parameters of the Extreme Component, covering frequency selection, probability estimation, and offset adjustment.

### 4.3 FREQUENCY-INFORMED OPTIMIZATION IN EFDIFF

In this section, we summarize how the sample prediction function $\hat{X}_0(X^{(k)}, k, \theta)$ is utilized in diffusion and present our overall optimization objective. In Eq. 4, we combine the Trend, Seasonality, and Extreme Components to form the prediction of the original clean sample $X^{(0)}$. During the denosing process, the update from $X^{(k)}$ to $X^{(k-1)}$ is given by:

$$X^{(k-1)} = \frac{1}{1-\bar{\alpha}_k}\Big(\sqrt{\bar{\alpha}_{k-1}}\,\beta_k\,\hat{\boldsymbol{X}}_{\boldsymbol{0}}(\boldsymbol{X^{(k)}}, \boldsymbol{k}, \boldsymbol{\theta}) + \sqrt{1-\beta_k}\,(1-\bar{\alpha}_{k-1})\,X^{(k)}\Big) + \tilde{\sigma}_k Z_k. \tag{18}$$

where $\beta_k$ is the noise variance at step $k$, $\alpha_k = 1 - \beta_k$ with $\bar{\alpha}_k = \prod_{i=1}^{k} \alpha_i$ denoting the cumulative product, $\tilde{\sigma}_k$ is the variance term in the reverse sampling process, and $\boldsymbol{Z}_k \sim \mathcal{N}(0, \boldsymbol{I})$ is Gaussian noise. In the final denoising step, $\hat{X}_0(X^{(k)}, k, \theta)$ is directly compared with the real sample $x^{(0)}$, and the network is optimized using a frequency-informed loss function. This objective integrates three complementary terms: (i) a distribution loss in the original time domain, (ii) a frequency-domain loss based on the Fourier representation, and (iii) an extreme-component loss defined over the extreme-contributing frequencies. The overall objective is formulated as:

$$\mathcal{L}_{\theta} = \mathbb{E}_{k, X^{(k)}, x^{(0)}}\Big[w_k\Big(\lambda_1 \big\|x^{(0)} - \hat{X}_0(X^{(k)}, k, \theta)\big\|_2^2 + \lambda_2\big\|\mathcal{DFT}(x^{(0)}) - \mathcal{DFT}(\hat{X}_0(X^{(k)}, k, \theta))\big\|_2^2$$
$$+ \lambda_3\, d_{\text{match}}\Big(\text{EFX}(x^{(0)}),\ \text{EFX}(\hat{X}_0(X^{(k)}, k, \theta))\Big)\Big)\Big], \quad w_t = \frac{\lambda\,\alpha_t\,(1-\bar{\alpha}_t)}{\beta_t^2}. \tag{19}$$

Here, $\mathcal{DFT}(\cdot)$ denotes the discrete Fourier transform, and $\text{EFX}(\cdot)$ is the Extreme-Frequency Extraction operator introduced in Section 4.1, which identifies the extreme-contributing frequencies from a given sample. The coefficients $\alpha_t$, $\bar{\alpha}_t$, and $\beta_t$ follow the standard diffusion definitions. The notation $\|\cdot\|_2^2$ indicates the squared $\ell_2$-norm. Finally, $d_{\text{match}}$ is a matching-based distance that measures the discrepancy between two sets of extreme-contributing frequencies.

## 5 EXPERIMENTS

We conducted extensive experiments to evaluate the performance of our EFDiff framework on five datasets across diverse scenarios using six metrics.

**Datasets** We evaluate our method on five real-world datasets from five different domains, including Economy (Stocks), Manufacturing (ETTh), Energy (HAE), HealthCare (fMRI), and Climate (Temperature). More detailed data description will be shown in the Appendix F.1.

**Metrics** We utilize six metrics to evaluate our model performance from different perspectives. In particular, we adopt Kullback–Leibler (KL) divergence Kullback & Leibler (1951), Jensen–Shannon

Table 1: Results on the extreme-value distribution of generated data, where all metrics are computed by comparison with the same extreme-value distribution of original data.

| Metrics | Methods | Stocks | ETTh | HAE | fMRI | Temperature |
|---|---|---|---|---|---|---|
| **KL Divergence** (Lower is better) | **TimeGAN** | 2.5304±0.3043 | 2.1225±0.1921 | 2.1334±0.2078 | 2.4162±0.2505 | 1.6329±0.1611 |
| | **beta-VAE** | 1.7783±0.2268 | 1.8004±0.1634 | 1.9052±0.2275 | 2.3588±0.3246 | 1.3336±0.1095 |
| | **TimeVAE** | 1.5325±0.1807 | 1.4287±0.1175 | 1.7745±0.2309 | 1.7633±0.1900 | 1.0524±0.0879 |
| | **Fourier-Flows** | 3.3261±0.4700 | 1.9943±0.2000 | 2.9518±0.3950 | 2.5811±0.3718 | 1.4358±0.1312 |
| | **Diffwave** | 1.1726±0.1304 | 1.5348±0.1727 | 1.5874±0.1577 | 0.9528±0.0809 | 0.9531±0.0806 |
| | **DiffTime** | 0.5218±0.0687 | 0.1131±0.0082 | 0.5153±0.0606 | 0.1136±0.0106 | 0.3718±0.0405 |
| | **FIDE** | 0.7849±0.0916 | 0.7635±0.0656 | 1.1263±0.1521 | 0.4529±0.0470 | 1.1542±0.1393 |
| | **Diffusion-TS** | 0.4666±0.0450 | 0.0356±0.0024 | 0.3463±0.0380 | 0.0665±0.0055 | 0.3957±0.0323 |
| | **EFDiff (ours)** | **0.2065±0.0165** | **0.0300±0.0021** | **0.3406±0.0307** | **0.0469±0.0038** | **0.2128±0.0170** |
| **JS Divergence** (Lower is better) | **TimeGAN** | 0.1508±0.0196 | 0.0784±0.0073 | 0.0883±0.0097 | 0.1071±0.0124 | 0.0431±0.0036 |
| | **beta-VAE** | 0.1359±0.0160 | 0.0771±0.0065 | 0.1458±0.0192 | 0.1132±0.0117 | 0.0631±0.0058 |
| | **TimeVAE** | 0.0847±0.0083 | 0.0643±0.0048 | 0.1422±0.0172 | 0.0873±0.0092 | 0.0587±0.0049 |
| | **Fourier-Flows** | 0.2114±0.0277 | 0.0952±0.0089 | 0.1252±0.0157 | 0.1354±0.0164 | 0.0367±0.0031 |
| | **Diffwave** | 0.1014±0.0109 | 0.0385±0.0029 | 0.9850±0.1270 | 0.0575±0.0056 | 0.0357±0.0027 |
| | **DiffTime** | 0.0566±0.0055 | 0.0098±0.0007 | 0.0793±0.0087 | 0.0124±0.0010 | 0.0187±0.0016 |
| | **FIDE** | 0.0973±0.0106 | 0.0218±0.0019 | 0.1135±0.0144 | 0.0251±0.0024 | 0.0285±0.0027 |
| | **Diffusion-TS** | 0.0520±0.0043 | 0.0085±0.0006 | 0.0767±0.0077 | 0.0081±0.0006 | 0.0178±0.0013 |
| | **EFDiff (ours)** | **0.0439±0.0031** | **0.0072±0.0004** | **0.0754±0.0060** | **0.0075±0.0005** | **0.0138±0.0010** |
| **Wasserstein Distance** (Lower is better) | **TimeGAN** | 0.0533±0.0049 | 0.0542±0.0045 | 0.1270±0.0139 | 0.0311±0.0027 | 0.0265±0.0024 |
| | **beta-VAE** | 0.1275±0.0161 | 0.1231±0.0123 | 0.1995±0.0233 | 0.0204±0.0017 | 0.0247±0.0019 |
| | **TimeVAE** | 0.0968±0.0117 | 0.1022±0.0106 | 0.2078±0.0237 | 0.0198±0.0017 | 0.0221±0.0018 |
| | **Fourier-Flows** | 0.1484±0.0199 | 0.1286±0.0121 | 0.2546±0.0328 | 0.0546±0.0068 | 0.0317±0.0029 |
| | **Diffwave** | 0.0524±0.0057 | 0.0511±0.0044 | 0.1435±0.0168 | 0.0246±0.0021 | 0.0199±0.0015 |
| | **DiffTime** | 0.0483±0.0051 | 0.0278±0.0021 | 0.1072±0.0115 | 0.0062±0.0005 | 0.0163±0.0013 |
| | **FIDE** | 0.0573±0.0057 | 0.0372±0.0033 | 0.1348±0.0146 | 0.0157±0.0014 | 0.0232±0.0019 |
| | **Diffusion-TS** | 0.0465±0.0043 | 0.0231±0.0019 | 0.0935±0.0093 | **0.0034±0.0002** | **0.0125±0.0009** |
| | **EFDiff (ours)** | **0.0412±0.0032** | **0.0208±0.0016** | **0.0902±0.0080** | 0.0044±0.0003 | 0.0155±0.0012 |
| **CRPS** (Lower is better) | **TimeGAN** | 0.1416±0.0187 | 0.1033±0.0083 | 0.1934±0.0234 | 0.0453±0.0044 | 0.1433±0.0119 |
| | **beta-VAE** | 0.1572±0.0191 | 0.1980±0.0217 | 0.1734±0.0189 | 0.0601±0.0068 | 0.1555±0.0129 |
| | **TimeVAE** | 0.1465±0.0159 | 0.1723±0.0176 | 0.1824±0.0208 | 0.0582±0.0064 | 0.1574±0.0130 |
| | **Fourier-Flows** | 0.1933±0.0271 | 0.2148±0.0237 | 0.1995±0.0274 | 0.0615±0.0076 | 0.1932±0.0201 |
| | **Diffwave** | 0.1624±0.0171 | 0.1792±0.0167 | 0.1623±0.0202 | 0.0473±0.0045 | 0.1538±0.0127 |
| | **DiffTime** | 0.1483±0.0144 | 0.1000±0.0074 | 0.1101±0.0098 | 0.0422±0.0031 | 0.1441±0.0122 |
| | **FIDE** | 0.1796±0.0196 | 0.1266±0.0107 | 0.1279±0.0127 | 0.0531±0.0050 | 0.1577±0.0127 |
| | **Diffusion-TS** | 0.1395±0.0122 | 0.0990±0.0073 | **0.1090±0.0081** | **0.0407±0.0029** | 0.1435±0.0105 |
| | **EFDiff (ours)** | **0.1393±0.0111** | **0.0981±0.0069** | 0.1098±0.0077 | 0.0410±0.0029 | **0.1425±0.0114** |

(JS) divergence Lin (2002), Wasserstein distance Villani et al. (2008), and Continuous Ranked Probability Score (CRPS) Gneiting & Raftery (2007), for *distributional fidelity evaluation*. We also utilize Predictive score Yoon et al. (2019) and Context-FID Jeha et al. (2022) to for *downstream quality evaluation* In general, the distributional metrics assess how well the generated data captures the extreme-value distributions, and all metrics are used to evaluate the overall generation quality, ensuring that improvements in extreme-value fidelity do not come at the expense of global data fidelity. A more detailed description of the metrics will be shown in the Appendix F.2.

**Baselines** We compare our proposed framework with eight representative generative models, including TimeGAN Yoon et al. (2019), beta-VAE Higgins et al. (2017), TimeVAE Desai et al. (2022), Fourier-Flows Alaa et al. (2021a), Diffwave Kong et al. (2021), DiffTime Coletta et al. (2023), Diffusion-TS Yuan & Qiao (2024), and FIDE Galib et al. (2024). Details are in the Appendix F.3.

## 5.1 Extreme-value Generation Results

To ensure fair comparison with prior work, we follow the evaluation protocol in recent studies such as FIDE Galib et al. (2024) and Diffusion-TS Yuan & Qiao (2024). We divide data into equal-length blocks of size 24 and apply the extreme-value extraction method PODT introduced in Section 3.1. Within each block, we select the top 10% of values to construct the extreme-value distribution of the data. Results on the extreme-value distribution of generated data are presented in Table 1. From this table, we found that our EFDiff consistently achieves the best results on all five datasets for KL

Divergence and JS Divergence. For Wasserstein Distance and CRPS, although it ranks second on two datasets, it achieves the best performance on the other three, demonstrating strong robustness. Among the four metrics, the most notable improvement is observed in KL Divergence. Since KL Divergence measures the information loss between generated and real distributions and is especially sensitive to the tail distribution, it is particularly effective in detecting whether a generative model fails to capture rare but critical events. This finding highlights the superior capability of our EFDiff in extreme-aware generation compared with existing methods.

Among the baselines, Diffusion-TS performs closest to EFDiff, mainly due to its refined disentanglement strategy. However, the lack of an explicit extreme-aware design limits its ability to model extreme events. In contrast, the recent method FIDE, despite employing a specialized Frequency-Inflated Conditional generation strategy, still performs poorly. This is because it does not consider the contribution of specific frequencies to extreme values.

## 5.2 OVERALL GENERATION RESULTS

We also evaluate the overall generation performance across six metrics, demonstrating that EFDiff not only improves extreme-value generation but also preserves overall data quality. Detailed results are provided in Appendix G.2. We present some visualizations of the overall distributions of our method and several baselines here. Figure 3a shows a 2D t-SNE Maaten & Hinton (2008) projection comparing EFDiff with a representative baseline, TimeGAN. The results indicate that EFDiff-generated data closely aligns with the original distribution, particularly in capturing fine-grained tail behaviors. In contrast, TimeGAN reproduces the main body of the distribution but fails to model finer tail structures and extreme events (e.g., outliers near the distribution boundary). Figure 3b further compares the probability density function (PDF) distributions of EFDiff and the strongest baseline, Diffusion-TS. While both methods yield similar overall distributions, the tail region (highlighted by the black box) reveals that Diffusion-TS underperforms in modeling the top 10% extreme values, whereas EFDiff aligns much more closely with the true tail behavior.

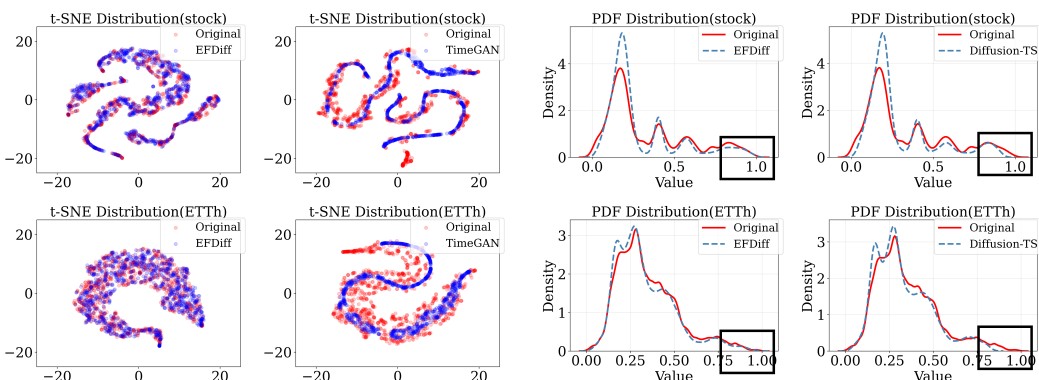

(a) t-SNE Distributions on the Stock/ETTh Datasets     (b) PDF Distributions on the Stock/ETTh Datasets

Figure 3: Overall Distributions of EFDiff and Baseline Models on the Stock and ETTh Datasets.

## 6 CONCLUSION

In this paper, we propose a novel frequency-informed diffusion framework called EFDiff for extreme-value time-series generation. Motivated by the observation that extreme values often arise from the superposition of multiple frequencies under specific phase alignments, we formulate extreme-value generation from a frequency-domain perspective, focusing on how to identify and exploit extreme-contributing frequencies. There are two key modules in EFDiff: (i) Extreme-Frequency Extraction (EFX), which captures overall extreme patterns as frequency combinations; and (ii) Extreme-Frequency Generation Enhancement (EFGEN), which leverages these combinations to enhance extreme structures during sampling. Extensive experiments on five real-world datasets across six metrics demonstrate that EFDiff not only achieves strong overall time-series generation quality but also significantly improves the fidelity of extreme-event generation, e.g., EFDiff improves 55.7% in KL Divergence on the Stocks dataset compared to the best baseline.

ETHICS STATEMENT

All authors have read and agree to abide by the ICLR Code of Ethics. This research uses only publicly available, de-identified datasets and does not involve human subjects, personally identifiable information, or sensitive private data. The study poses no foreseeable risk of harm to individuals or groups, and all experiments were conducted in compliance with applicable laws and institutional policies. No conflicts of interest or external sponsorship that could bias the results are present. We take full responsibility for the integrity of the data, the analyses, and the conclusions presented in this paper.

REPRODICIBILITY STATEMENT

We have taken extensive steps to ensure the reproducibility of our work. All model architectures, training procedures, and hyperparameter settings are described in detail in Section 4 and Section 5 of the paper. More detailed information on experiment datasets, metrics, and baselines is described in Appendix F. To improve reproducibility, we release an anonymous **GitHub: https://anonymous.4open.science/r/EFDiff** repository containing the full source code, datasets, configuration files, and scripts for data preparation and model training as part of the supplementary materials during the review process. These resources together enable independent researchers to reproduce all reported results and figures.

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

## A    The Use of Large Language Models

Large language models (LLMs) were not involved in the research ideation, methodology design, data analysis, or result interpretation. LLMs were only used for language polishing and minor grammar/style suggestions after the manuscript was fully drafted by the authors. All conceptual content and scientific contributions are entirely the authors' own, and the authors take full responsibility for the final text.

## B    Related Work

### B.1    Time Series Generation

Early work on time-series generation adapted general-purpose generative models to sequential data, with two families proving especially influential: GAN-based adversarial training and VAE-based variational autoencoding. TimeGAN Yoon et al. (2019) combines adversarial learning with a supervised sequence encoder to preserve stepwise dependencies and is widely used as a baseline. $\beta$-VAE Higgins et al. (2017) augments the VAE objective with a disentanglement coefficient for interpretable latent factors and controllable synthesis, while TimeVAE Desai et al. (2021) adapts the framework to multivariate sequences with sequence-aware components that improve reconstruction and interpretability. Fourier Flows Alaa et al. (2021b) integrate Fourier-domain parameterizations with normalizing flows to model complex temporal spectra and variable sampling more faithfully.

In recent years, diffusion models have emerged as the state-of-the-art generative paradigm across modalities, including image, video, audio, text, and increasingly time series. Compared with GAN-based Yoon et al. (2019) and VAE-based methods Naiman et al. (2024), diffusion models can better capture complex temporal dependencies and generate sequences that are both realistic and diverse. Unlike Transformer-based LLM approaches Chen et al. (2024) that rely on stepwise conditional predictions, diffusion models preserve fidelity to the overall distribution, enabling them to capture rare events from a global perspective. Within time series, DiffWave Kong et al. (2021) popularized

diffusion in audio, paving the way for sequential signals. Beyond task-specific conditioning, TS-Diff Kollovieh et al. (2023) trains an unconditional prior and uses inference-time self-guidance to steer samples, while Diffusion-TS Yuan & Qiao (2024) integrates trend–seasonality decomposition with diffusion to improve fidelity and interpretability. DiffTime Coletta et al. (2023) incorporates numerical and physical constraints for long sequences. Despite these advances, diffusion-based models often neglect the extreme-value regime, failing to capture rare yet consequential events that are critical in many applications.

## B.2 EXTREME-AWARE GENERATION

Extreme-aware generation seeks to faithfully model the high-impact, low-probability regions of data distributions that conventional generators often fail to capture. ExGAN Bhatia et al. (2021) directs learning toward exceedances by up-weighting samples beyond a learned threshold, providing stronger gradient signals for rare regions without degrading typical-case realism. ParetoGAN Huster et al. (2021) extends adversarial generation to heavy-tailed targets by explicitly parameterizing the tail with a generalized Pareto form, enabling the model to match tail indices and synthesize realistic extremes. EV-GAN Allouche et al. (2022b) formalizes exceedance modeling with EVT-consistent objectives in ReLU networks, offering principled control above high thresholds and improved tail sample quality. FIDE Galib et al. (2024) enhances diffusion-based time-series synthesis of rare events by inflating frequency components and reweighting conditional objectives so that sharp and infrequent patterns receive sufficient learning mass while preserving average-case fidelity.

Unlike these methods, which primarily preserve long-tail distributions in the time domain, our proposed EFDiff adopts a frequency-domain perspective that is more effective for modeling extreme-value distributions. In time-series analysis, techniques such as the Fourier transform Percival & Walden (1993) and seasonal–trend decomposition Liu et al. (2023) underscore the frequency domain as a crucial perspective, offering powerful opportunities to study extremes. Although a recent study Galib et al. (2024) made an initial attempt at frequency-based extreme generation, it neither provides quantitative analysis under inflated frequencies nor moves beyond conditioning on a fixed GEV distribution, thereby overlooking richer extreme-value patterns. Consequently, exploring the underlying structures of extremes in the frequency domain and leveraging frequency-informed information to facilitate extreme-value generation remains an open challenge.

## C DENOISING DIFFUSION PROBABILISTIC MODELS

Among diffusion-based generative models, the *Denoising Diffusion Probabilistic Model (DDPM)* Ho et al. (2020) is the most widely used discrete-time formulation. It casts generation as the inversion of a simple Gaussian noising process: a fixed forward Markov chain that gradually corrupts data and a learned reverse process that denoises step by step, yielding stable training, a tractable likelihood objective, and high-fidelity samples across modalities.

The forward diffusion process is a non-trainable Markov chain that transforms real sample $\mathbf{x}^0$ into latent variables $\mathbf{x}^1, \ldots, \mathbf{x}^N$, which can be represented as:

$$q(\mathbf{x}^n|\mathbf{x}^{n-1}) = \mathcal{N}(\mathbf{x}^n; \sqrt{1-\beta_n}\mathbf{x}^{n-1}, \beta_n\mathbf{I}), \tag{20}$$

where $\beta_1, \ldots, \beta_N \in (0,1)$ is a pre-determined variance schedule. $\mathbf{x}^n$ can be directly sampled from $\mathbf{x}^0$ with $q(\mathbf{x}^n|\mathbf{x}^0) = \mathcal{N}(\mathbf{x}^n; \sqrt{\overline{\alpha}_n}\mathbf{x}^0, (1-\overline{\alpha}_n)\mathbf{I})$, where $\alpha_n = 1-\beta_n$ and $\overline{\alpha}_n = \prod_{n=1}^{N}\alpha_n$. With the application of the reparameterization, $\mathbf{x}^n$ can be represented as $\mathbf{x}^n = \sqrt{\overline{\alpha}_n}\mathbf{x}^0 + \sqrt{1-\overline{\alpha}_n}\boldsymbol{\epsilon}$, where $\boldsymbol{\epsilon} \sim \mathcal{N}(0,\mathbf{I})$. The backward diffusion process is a trainable Markov chain that aims to recover $\mathbf{x}^0$ from $\mathbf{x}^N$, which can be formulated as:

$$p_\theta(\mathbf{x}^{n-1}|\mathbf{x}^n) = \mathcal{N}(\mathbf{x}^{n-1}; \mu_\theta(\mathbf{x}^n, n), \sigma_\theta(\mathbf{x}^n, n)\mathbf{I}), \tag{21}$$

where $\mu_\theta(\mathbf{x}^n, n)$ and $\sigma_\theta(\mathbf{x}^n, n)$ are the mean and variance predicted normally by a neural network parameterized by $\theta$. The loss function is formalized as:

$$\mathbb{E}_{\mathbf{x}^0, \boldsymbol{\epsilon}}\left[\frac{\beta_n^2}{2\Sigma_\theta\alpha_n(1-\overline{\alpha}_n)}\left\|\boldsymbol{\epsilon} - \epsilon_\theta\left(\sqrt{\overline{\alpha}_n}\mathbf{x}^0 + \sqrt{1-\overline{\alpha}_n}\boldsymbol{\epsilon}, n\right)\right\|^2\right], \tag{22}$$

where $\epsilon_\theta$ is a neural network for predicting sampled $\boldsymbol{\epsilon} \sim \mathcal{N}(0,\mathbf{I})$. After trained, trajectory genera-

tion is conducted by progressively sampling $\mathbf{x}^{n-1}$ from distribution $p_\theta(\mathbf{x}^{n-1}|\mathbf{x}^n)$ until reach $\mathbf{x}^0$ by computing:

$$\mathbf{x}^{n-1} = \frac{1}{\sqrt{\alpha_n}}\left(\mathbf{x}^n - \frac{\beta_n}{\sqrt{1-\alpha_n}}\epsilon_\theta(\mathbf{x}^n, n)\right) + \sqrt{\Sigma_\theta}\mathbf{z}, \tag{23}$$

where $\mathbf{z} \sim \mathcal{N}(\mathbf{0}, \mathbf{I})$ for $n \in [2, N]$, and $\mathbf{z} = \mathbf{0}$ when $n = 1$.

## D  FOURIER TRANSFORM

The Fourier transform (FT) is a fundamental integral transform that represents a time- or spatial-domain signal as a superposition of complex exponentials (frequencies). For a function $f(t) \in L^1(\mathbb{R})$, the continuous Fourier transform and its inverse are defined as:

$$\mathcal{F}\{f\}(\omega) = F(\omega) = \int_{-\infty}^{\infty} f(t)\,e^{-i\omega t}\,\mathrm{d}t, \tag{24}$$

$$\mathcal{F}^{-1}\{F\}(t) = f(t) = \frac{1}{2\pi}\int_{-\infty}^{\infty} F(\omega)\,e^{i\omega t}\,\mathrm{d}\omega. \tag{25}$$

For discrete, finite-length signals $x[n]$ of length $N$, the Discrete Fourier Transform (DFT) and inverse DFT (IDFT) are given by:

$$X[k] = \sum_{n=0}^{N-1} x[n]\,e^{-i2\pi kn/N}, \qquad k = 0, \ldots, N-1, \tag{26}$$

$$x[n] = \frac{1}{N}\sum_{k=0}^{N-1} X[k]\,e^{i2\pi kn/N}, \qquad n = 0, \ldots, N-1. \tag{27}$$

Recent works have explored how Fourier-domain analysis or Fourier-based modifications to the diffusion process can improve generation quality, especially for high-frequency details. One key observation is that in standard denoising diffusion probabilistic models (DDPMs), as mentioned above, the forward noising process with additive white Gaussian noise corrupts high-frequency components much faster than low-frequency ones in terms of signal-to-noise ratio (SNR), leading to delayed recovery of fine details during sampling Falck et al. (2025). To formalize, let $x_0$ be an image and consider the standard forward process at time $t$:

$$x_t = \alpha_t x_0 + \sigma_t \epsilon, \qquad \epsilon \sim \mathcal{N}(0, I), \tag{28}$$

where $\alpha_t$ and $\sigma_t$ are scalar schedule parameters. The SNR for a frequency component $\omega$ at time $t$ can be expressed as:

$$\mathrm{SNR}_t(\omega) = \frac{|\alpha_t \mathcal{F}(x_0)(\omega)|^2}{\sigma_t^2\,\mathrm{Var}(\mathcal{F}(\epsilon)(\omega))}. \tag{29}$$

High-$\omega$ components suffer greater relative degradation due to their lower magnitude in natural images since $\mathrm{Var}(\mathcal{F}(\epsilon)(\omega))$ is typically uniform across frequencies for white noise.

## E  TREND AND SEASONALITY COMPONENT

**Trend Component:** For the trend component, we rely on low-frequency signals to model slow, long-term variations. We avoid using the Fourier transform for trend modeling, since extended trends may yield incomplete or distorted periodic representations. Instead, inspired by Oreshkin et al. (2020); Desai et al. (2022), we adopt a polynomial regressor to represent the trend as:

$$trend(X_{enc}^{(k)}, \theta_{tr}) = \phi(t)^\top\big(W_{tr}\,h(X_{enc}^{(k)}; a_{tr}) + b_{tr}\big), \tag{30}$$

where $X_{enc}^{(k)}$ is the noisy time-series sequence at diffusion step $k$ after encoding, $t$ is the time index, and $\phi(t) = [1, t, t^2, \ldots, t^p]^\top$ is the polynomial basis of degree $p$. The function $h(X_{enc}^{(k)}; a_{tr})$ denotes a neural feature extractor parameterized by $a_{tr}$, $W_{tr}$ is a learnable projection matrix that

maps the extracted features into polynomial coefficients, and $b_{tr}$ is a bias vector. Together, $W_{tr}$, $a_{tr}$, and $b_{tr}$ constitute $\theta_{tr}$. In this formulation, the network part $W_{tr}\, h(X_{enc}^{(k)}; a_{tr}) + b_{tr}$ produces polynomial coefficients, while the basis vector $\phi(t)$ encodes temporal structure, and their inner product yields the trend value. Since the trend mainly characterizes low-frequency information, $p$ is usually set to a small value such as 3.

**Seasonality Component:** We employ the discrete Fourier transform (DFT) to decompose $X_{enc}^{(k)}$ and identify dominant frequencies for reconstructing $\hat{X}^{(0)}$. These dominant periodic patterns typically correspond to the frequency components with the largest magnitudes in the Fourier domain. Accordingly, we select the Top-$K$ frequencies with the highest amplitudes and recombine them through the inverse discrete Fourier transform (IDFT). The formulation is given by:

$$seasonality\left(X_{enc}^{(k)}, \theta_{se}\right) = \mathcal{IDFT}\Big(\mathcal{M}_K\big(\mathcal{DFT}(X_{enc}^{(k)}), \theta_{se}\big)\Big), \tag{31}$$

where $\mathcal{DFT}$ and $\mathcal{IDFT}$ denote the discrete Fourier transform and its inverse, $\mathcal{M}_K(\cdot, \theta_{se})$ is a masking and reweighting operator that preserves the Top-$K$ frequencies with the largest magnitudes along with their conjugate-symmetric counterparts, and $\theta_{se}$ denotes a learnable Transformer network applied to the selected frequencies, which models their interactions and refines their contributions to better reconstruct the seasonal component.

# F  IMPLEMENTATION DETAILS

## F.1  DATASETS

We evaluate our method on five real-world datasets, which are summarized in Table 2.

Table 2: Dataset Statistics.

| Datasets | # of Samples | Category | Link |
|---|---|---|---|
| Stocks | 3685 | Economy | https://finance.yahoo.com/quote/GOOG |
| ETTh | 17420 | Manufacturing | https://github.com/zhouhaoyi/ETDataset |
| HAE | 19735 | Energy | https://archive.ics.uci.edu/ml/datasets |
| fMRI | 10000 | HealthCare | https://www.fmrib.ox.ac.uk/datasets |
| Temperature | 96432 | Climate | https://cds.climate.copernicus.eu/ |

- **Stocks dataset** contains stock price records from Yahoo Finance spanning 2004 to 2019, where we use the adjusted closing price.

- **ETTh dataset** consists of electricity transformer records collected from July 2016 to July 2018, including multiple power-related features, among which we use oil temperature.

- **HAE dataset** contains household energy consumption records collected over 4.5 months at 10-minute intervals, incorporating indoor climate, weather, and appliance usage features, where we focus on the energy consumption of appliances.

- **fMRI dataset** provides realistic simulated time series designed for evaluating brain network modeling methods under diverse network structures, experimental protocols, and potential confounds. From this dataset, we select 10,000 testing samples from a single subject.

- **Temperature dataset** is derived from the ERA5, which provides hourly global reanalysis data on single levels from 1940 to the present, including atmospheric, ocean-wave, and land-surface variables. Specifically, we use ten years of temperature data from Verkhoyansk in Russia, which is recognized as one of the locations with the most extreme temperatures worldwide.

## F.2  METRICS

In this part, we introduce six metrics, comprising four metrics for evaluating distributional properties as well as temporal and feature dependencies, and two metrics for assessing generation quality through downstream task evaluation.

For **distributional evaluation**, we adopt the following four metrics:

- **Kullback–Leibler (KL) divergence** Kullback & Leibler (1951), which measures the information discrepancy of one distribution relative to another in an asymmetric manner.

- **Jensen–Shannon (JS) divergence** Lin (2002), a symmetric and bounded variant of KL divergence that evaluates the similarity between two distributions.

- **Wasserstein distance** Villani et al. (2008), which quantifies the minimal transport cost between the cumulative distribution functions of two distributions.

- **Continuous Ranked Probability Score (CRPS)** Gneiting & Raftery (2007), which captures the overall difference between the cumulative distribution function of a predictive distribution and the empirical observation.

For **downstream quality evaluation**, we consider two complementary metrics:

- **Predictive score** Yoon et al. (2019), which evaluates the utility of synthetic data in next-step forecasting tasks based on the Train-on-Synthetic-Test-on-Real (TSTR) protocol.

- **Context-FID** Jeha et al. (2022), which compares local contextual representations to measure the fidelity of generated time series.

### F.3 BASELINES

We compare our proposed framework with eight representative generative models spanning diverse paradigms, including VAE, GAN, and Diffusion.

- **TimeGAN** Yoon et al. (2019), a GAN-based model specifically designed for time-series generation, which integrates supervised and unsupervised objectives to capture temporal dynamics.

- **beta-VAE** Higgins et al. (2017) extends the variational autoencoder by introducing a disentanglement coefficient, which encourages the learning of interpretable latent representations.

- **TimeVAE** Desai et al. (2022) adapts the VAE framework to time series by incorporating sequence-aware architectures that improve interpretability and reconstruction quality.

- **Fourier-Flows** Alaa et al. (2021a) is a flow-based approach that combines Fourier features with normalizing flows to model complex temporal distributions.

- **Diffwave** Kong et al. (2021), originally proposed for raw audio generation, employs a diffusion probabilistic process and has been adapted to time-series synthesis.

- **DiffTime** Coletta et al. (2023) extends diffusion models to long sequences by explicitly exploiting temporal structures.

- **Diffusion-TS** Yuan & Qiao (2024) introduces a transformer-based diffusion framework tailored for time-series generation.

- **FIDE** Galib et al. (2024) is an extreme-aware framework that enhances conditional diffusion models with inflated frequency representations for improved generation quality.

### F.4 EXPERIMENT SETUP AND PARAMETER SETTING

Our experiments are conducted using Python 3.9.21 with implementations based on the PyTorch deep learning framework Paszke et al. (2019). All models are trained and evaluated on an Amazon EC2 g5.8xlarge instance equipped with a single NVIDIA A10G Tensor Core GPU with 24 GB memory. The detailed parameter setting is shown in Table 3 and Table 4. More details can be seen in the code Config page.

Table 3: Experimental configurations for model, training, and data.

| Parameter | Value / Description |
|---|---|
| **Model settings** | |
| Sequence length | 24 time steps |
| Feature dimension | 1 |
| Number of encoder layers | 2 |
| Number of decoder layers | 2 |
| Model hidden dimension | 64 |
| Number of attention heads | 4 |
| MLP hidden expansion factor | 4 |
| Diffusion timesteps | 500 |
| Sampling timesteps | 500 |
| Loss function | L1 loss |
| Beta schedule | Cosine |
| Attention dropout | 0.0 |
| Residual dropout | 0.0 |
| Convolution kernel size | 1 |
| Convolution padding size | 0 |
| **Training settings** | |
| Base learning rate | $1.0 \times 10^{-5}$ |
| Maximum epochs | 10,000 |
| Gradient accumulation steps | 2 |
| EMA decay rate | 0.995 (update every 10 steps) |
| Scheduler reduction factor | 0.5 |
| Scheduler patience | 2000 epochs |
| Warmup learning rate | $8.0 \times 10^{-4}$ |
| Warmup steps | 500 |
| Minimum learning rate | $1.0 \times 10^{-5}$ |
| **Data settings** | |
| Batch size | 64 |
| Sample size per iteration | 256 |
| Test sampling coefficient | $1.0 \times 10^{-2}$ |
| Test step size | $5.0 \times 10^{-2}$ |
| Test sampling steps | 200 |

## G MORE RESULTS

### G.1 ALBATION STUDY

In this section, we investigate how the three frequency-based extreme-contribution measurements in our core design EFX (introduced in Section 4.1) affect extreme-value generation. To evaluate their individual roles, we construct three ablated variants of our method: EFDiff-$PLV$ denotes EFDiff without the PLV measurement, while EFDiff-$C_w$ and EFDiff-$L_w$ are obtained by removing the $C_w$ and $L_w$ measurements, respectively. As shown in Table 5, removing any single measurement reduces the performance of EFDiff in extreme-value generation. The effect of $L_w$ is relatively minor, as it only reflects the incremental contribution of a frequency relative to the background. In contrast, $C_w$ and PLV have a much greater impact, since they directly capture the strength and phase alignment of frequency combinations driving extreme events.

### G.2 OVERALL GENERATION RESULTS

In this part, we also evaluate the overall generation results, showing that our EFDiff not only enhances extreme-value generation but also maintains the overall quality of generation. For the overall distribution, we use six metrics: the first four distributional metrics measure generation fidelity, while the remaining two assess the utility of the generated data via downstream tasks (prediction and representation learning). The complete results are reported in Table 6.

Table 4: Extreme training and extraction settings.

| Parameter | Value / Description |
|---|---|
| **Extreme training settings** | |
| Extreme training epochs | 20 |
| Extreme training batch size | 64 |
| Extreme training learning rate | $3.0 \times 10^{-4}$ |
| Extreme training weight decay | $1.0 \times 10^{-4}$ |
| Extreme training hidden dimension | 128 |
| Extreme diffusion $\beta$ range | $[1.0 \times 10^{-4}, 2.0 \times 10^{-2}]$ |
| Delta ($\delta$) | 2 |
| **Top-$k$** | **5** |
| Top-$k$ margin | 0.4 |
| Top-$k$ loss weight ($\lambda_{topk}$) | $2.0 \times 10^{-3}$ |
| Positive class weight | 1.6 |
| Late bias factor ($\gamma$) | 5.0 |
| Stability constant ($\varepsilon$) | $1.0 \times 10^{-8}$ |
| Random seed | 42 |
| Device | CUDA |
| **Extreme extraction settings** | |
| Proportion for extreme extraction ($p$) | 0.10 |
| Mean-ratio threshold factor ($\alpha_{mean}$) | 0.05 |
| Standard deviation threshold factor ($k_\sigma$) | 1.5 |
| Median absolute deviation threshold factor ($k_{mad}$) | 2.5 |
| Neighborhood parameter ($\rho_{near}$) | 0.02 |
| Dilation parameter ($\rho_{dil}$) | 0.02 |
| Trend estimation method | Linear |

Table 5: Ablation Study on the extreme-value distribution of generated data, where all metrics are computed by comparison with the same distribution of original data.

| Metrics | Methods | Stocks | ETTh | HAE | fMRI | Temperature |
|---|---|---|---|---|---|---|
| **KL Divergence** (Lower is better) | EFDiff-$PLV$ | 0.3129±0.0310 | 0.0344±0.0034 | 0.4005±0.0365 | 0.1113±0.0109 | 0.2757±0.0265 |
| | EFDiff-$C_w$ | 0.4788±0.0485 | 0.0344±0.0035 | 0.4316±0.0429 | 0.1295±0.0132 | 0.3931±0.0388 |
| | EFDiff-$L_w$ | 0.2524±0.0243 | 0.0303±0.0029 | 0.3452±0.0354 | 0.0512±0.0051 | 0.2343±0.0236 |
| | **EFDiff** | **0.2065±0.0165** | **0.0300±0.0021** | **0.3406±0.0307** | **0.0469±0.0038** | **0.2128±0.0170** |
| **JS Divergence** (Lower is better) | EFDiff-$PLV$ | 0.0481±0.0039 | 0.0080±0.0010 | 0.0765±0.0059 | 0.0081±0.0006 | 0.0183±0.0015 |
| | EFDiff-$C_w$ | 0.0533±0.0044 | 0.0094±0.0008 | 0.0787±0.0066 | 0.0082±0.0007 | 0.0192±0.0016 |
| | EFDiff-$L_w$ | 0.0460±0.0040 | 0.0079±0.0007 | 0.0754±0.0068 | 0.0078±0.0007 | 0.0148±0.0013 |
| | **EFDiff** | **0.0439±0.0031** | **0.0072±0.0004** | **0.0754±0.0060** | **0.0075±0.0005** | **0.0138±0.0010** |
| **Wasserstein Distance** (Lower is better) | EFDiff-$PLV$ | 0.0483±0.0040 | 0.0232±0.0021 | 0.0933±0.0079 | 0.0074±0.0006 | 0.0176±0.0015 |
| | EFDiff-$C_w$ | 0.0512±0.0043 | 0.0255±0.0022 | 0.0942±0.0080 | 0.0080±0.0010 | 0.0203±0.0017 |
| | EFDiff-$L_w$ | 0.0447±0.0038 | 0.0211±0.0018 | 0.0919±0.0080 | 0.0057±0.0005 | 0.0162±0.0014 |
| | **EFDiff** | **0.0412±0.0032** | **0.0208±0.0016** | **0.0902±0.0080** | **0.0044±0.0003** | **0.0155±0.0012** |
| **CRPS** (Lower is better) | EFDiff-$PLV$ | 0.1408±0.0117 | 0.1037±0.0086 | 0.1143±0.0098 | 0.0499±0.0043 | 0.1442±0.0125 |
| | EFDiff-$C_w$ | 0.1402±0.0137 | 0.1042±0.0100 | 0.1122±0.0113 | 0.0510±0.0050 | 0.1458±0.0143 |
| | EFDiff-$L_w$ | 0.1399±0.0127 | **0.0978±0.0084** | 0.1120±0.0100 | 0.0426±0.0036 | 0.1436±0.0124 |
| | **EFDiff** | **0.1393±0.0111** | 0.0981±0.0069 | **0.1098±0.0077** | **0.0410±0.0029** | **0.1425±0.0114** |

From Table 6, we found that our EFDiff achieves the best performance on most metrics and datasets. Specifically, we achieve the best results on the Stocks, ETTh, fMRI, and Temperature datasets, while only performing weaker on HAE. This is because the HAE dataset logs room energy consumption every ten minutes and contains many zeros and fixed integer-valued observations. In such zero-inflated, discrete settings, diffusion-based models, which rely on continuous Gaussian perturbations and continuous-density objectives, tend to blur the point mass at zero and other discrete spikes, whereas GAN-based and VAE-based approaches can better accommodate this structure; for example, VAEs can use discrete or zero-inflated likelihoods, and GANs can rely on implicit distribution modeling that avoids explicit likelihood assumptions.

Table 6: Results on the overall distribution of generated data, where all metrics are computed by comparison with the same distribution of original data (bold indicates best performance, underline indicates second best).

| Metrics | Methods | Stocks | ETTh | HAE | fMRI | Temperature |
|---|---|---|---|---|---|---|
| **KL Divergence** (Lower is better) | TimeGAN | 1.9692±0.186 | 1.7851±0.149 | 0.3015±0.029 | 0.3626±0.034 | 0.9497±0.087 |
| | beta-VAE | 1.9009±0.223 | 1.6458±0.168 | 0.4579±0.064 | 0.4982±0.058 | 0.7351±0.076 |
| | TimeVAE | 1.7821±0.170 | 1.1782±0.112 | 0.4050±0.045 | 0.3524±0.033 | 0.6658±0.075 |
| | Fourier-Flows | 2.5635±0.354 | 3.6871±0.447 | 0.6325±0.090 | 0.7841±0.114 | 1.1198±0.136 |
| | Diffwave | 1.1892±0.127 | 0.8827±0.075 | 0.3483±0.043 | 0.3383±0.043 | 0.3896±0.044 |
| | DiffTime | 0.3045±0.035 | 0.0820±0.0069 | 0.1999±0.022 | 0.0254±0.0034 | 0.1576±0.018 |
| | FIDE | 1.1177±0.133 | 0.9525±0.093 | 0.2287±0.029 | 0.0191±0.0023 | 0.3548±0.040 |
| | Diffusion-TS | 0.3205±0.036 | 0.0158±0.0012 | **0.1872±0.019** | 0.0076±0.0010 | 0.1317±0.014 |
| | EFDiff (ours) | **0.1460±0.013** | **0.0125±0.0010** | 0.1975±0.026 | **0.0066±0.0007** | **0.0721±0.007** |
| **JS Divergence** (Lower is better) | TimeGAN | 0.1219±0.0118 | 0.0702±0.0053 | **0.0369±0.0033** | 0.0228±0.0020 | 0.0498±0.0039 |
| | beta-VAE | 0.1589±0.0192 | 0.0791±0.0068 | 0.0470±0.0066 | 0.0247±0.0026 | 0.0400±0.0043 |
| | TimeVAE | 0.1320±0.0113 | 0.0630±0.0048 | 0.0442±0.0046 | 0.0219±0.0019 | 0.0389±0.0044 |
| | Fourier-Flows | 0.2236±0.0317 | 0.0855±0.0073 | 0.0643±0.0091 | 0.0252±0.0025 | 0.0510±0.0057 |
| | Diffwave | 0.0732±0.0067 | 0.0118±0.0011 | 0.0571±0.0074 | 0.0166±0.0014 | 0.0256±0.0027 |
| | DiffTime | 0.0472±0.0047 | 0.0039±0.0004 | 0.0488±0.0058 | 0.0025±0.0003 | 0.0171±0.0015 |
| | FIDE | 0.0997±0.0115 | 0.0165±0.0015 | 0.0550±0.0074 | 0.0178±0.0017 | 0.0288±0.0032 |
| | Diffusion-TS | 0.0370±0.0039 | 0.0038±0.0004 | 0.0394±0.0041 | 0.0013±0.0001 | 0.0133±0.0014 |
| | EFDiff (ours) | **0.0313±0.0031** | **0.0030±0.0003** | 0.0418±0.0054 | **0.0011±0.0001** | **0.0102±0.0011** |
| **Wasserstein Distance** (Lower is better) | TimeGAN | 0.0524±0.0053 | 0.0378±0.0033 | **0.0282±0.0026** | 0.0203±0.0020 | 0.0246±0.0021 |
| | beta-VAE | 0.0782±0.0109 | 0.0412±0.0039 | 0.0356±0.0051 | 0.0387±0.0049 | 0.0244±0.0020 |
| | TimeVAE | 0.0778±0.0069 | 0.0376±0.0025 | 0.0331±0.0031 | 0.0258±0.0020 | 0.0275±0.0030 |
| | Fourier-Flows | 0.1245±0.0159 | 0.0540±0.0041 | 0.0621±0.0087 | 0.0331±0.0040 | 0.0339±0.0029 |
| | Diffwave | 0.0652±0.0076 | 0.0393±0.0030 | 0.0457±0.0063 | 0.0098±0.0009 | 0.0245±0.0019 |
| | DiffTime | 0.0451±0.0046 | 0.0199±0.0014 | 0.0432±0.0052 | 0.0040±0.0005 | 0.0138±0.0013 |
| | FIDE | 0.0483±0.0061 | 0.0247±0.0021 | 0.0388±0.0046 | 0.0072±0.0007 | 0.0255±0.0027 |
| | Diffusion-TS | 0.0447±0.0052 | 0.0171±0.0017 | 0.0305±0.0039 | **0.0035±0.0003** | 0.0117±0.0009 |
| | EFDiff (ours) | **0.0397±0.0036** | **0.0141±0.0011** | 0.0305±0.0037 | 0.0040±0.0004 | **0.0108±0.0009** |
| **CRPS** (Lower is better) | TimeGAN | 0.1377±0.012 | 0.0958±0.009 | **0.0382±0.0046** | 0.0768±0.0064 | 0.1404±0.011 |
| | beta-VAE | 0.1412±0.016 | 0.1045±0.009 | 0.0410±0.0050 | 0.0873±0.010 | 0.1633±0.018 |
| | TimeVAE | 0.1369±0.013 | 0.0972±0.008 | 0.0397±0.0045 | 0.0983±0.013 | 0.1521±0.017 |
| | Fourier-Flows | 0.1920±0.024 | 0.1244±0.014 | 0.0691±0.0102 | 0.1129±0.015 | 0.2371±0.030 |
| | Diffwave | 0.1583±0.015 | 0.0998±0.009 | 0.0580±0.0071 | 0.0852±0.009 | 0.1587±0.016 |
| | DiffTime | 0.1358±0.013 | 0.0951±0.008 | 0.0531±0.0068 | 0.0769±0.007 | 0.1492±0.017 |
| | FIDE | 0.1437±0.017 | 0.1183±0.011 | 0.0455±0.0062 | 0.0788±0.009 | 0.1588±0.018 |
| | Diffusion-TS | **0.1355±0.012** | 0.0944±0.007 | 0.0412±0.0050 | 0.0768±0.0069 | **0.1399±0.011** |
| | EFDiff (ours) | 0.1361±0.011 | **0.0937±0.007** | 0.0410±0.0049 | **0.0757±0.0069** | 0.1403±0.011 |
| **Predictive Score** (Lower is better) | TimeGAN | 0.1203±0.011 | 0.0195±0.0020 | **0.0474±0.0060** | 0.1064±0.011 | 0.0184±0.0015 |
| | beta-VAE | 0.1995±0.026 | 0.0259±0.0024 | 0.0581±0.0070 | 0.1130±0.013 | 0.0248±0.0029 |
| | TimeVAE | 0.1816±0.020 | 0.0233±0.0020 | 0.0538±0.0066 | 0.1049±0.012 | 0.0247±0.0027 |
| | Fourier-Flows | 0.2130±0.028 | 0.0251±0.0026 | 0.1012±0.0132 | 0.1195±0.015 | 0.0291±0.0035 |
| | Diffwave | 0.1792±0.017 | 0.0235±0.0023 | 0.0783±0.0100 | 0.1032±0.012 | 0.0284±0.0029 |
| | DiffTime | 0.0983±0.010 | 0.0194±0.0018 | 0.0631±0.0083 | 0.1004±0.011 | 0.0221±0.0022 |
| | FIDE | 0.1897±0.021 | 0.0213±0.0019 | 0.0591±0.0074 | 0.1094±0.014 | 0.0235±0.0024 |
| | Diffusion-TS | 0.1160±0.011 | 0.0182±0.0015 | 0.0485±0.0061 | **0.0955±0.0100** | 0.0188±0.0019 |
| | EFDiff (ours) | **0.0201±0.0024** | **0.0156±0.0014** | 0.0481±0.0056 | 0.0959±0.0110 | **0.0180±0.0016** |
| **Context FID** (Lower is better) | TimeGAN | 0.1349±0.013 | 0.1799±0.018 | **0.1519±0.015** | 0.2015±0.021 | 0.0749±0.0059 |
| | beta-VAE | 0.1618±0.019 | 0.1390±0.015 | 0.1649±0.021 | 0.2254±0.029 | 0.0837±0.0086 |
| | TimeVAE | 0.1570±0.015 | 0.1342±0.012 | 0.1572±0.021 | 0.1963±0.023 | 0.0559±0.0049 |
| | Fourier-Flows | 0.2183±0.031 | 0.2286±0.030 | 0.3055±0.048 | 0.3053±0.038 | 0.1782±0.021 |
| | Diffwave | 0.1392±0.013 | 0.0539±0.0047 | 0.2791±0.039 | 0.1264±0.014 | 0.0225±0.0021 |
| | DiffTime | 0.0854±0.0081 | 0.0305±0.0030 | 0.2462±0.034 | 0.0088±0.0010 | 0.0186±0.0021 |
| | FIDE | 0.1077±0.011 | 0.0791±0.0070 | 0.2877±0.042 | 0.1371±0.015 | 0.0344±0.0036 |
| | Diffusion-TS | 0.0794±0.0072 | 0.0279±0.0025 | 0.2040±0.027 | 0.0081±0.0009 | **0.0080±0.0008** |
| | EFDiff (ours) | **0.0479±0.0045** | **0.0207±0.0019** | 0.2341±0.035 | **0.0075±0.0009** | 0.0099±0.0011 |

## G.3 MORE VISUALIZATIONS

In this part, we provide additional overall distribution visualizations using 2D t-SNE Maaten & Hinton (2008) and PDF distribution. We select two classic baselines, TimeGAN and Diffusion-TS for representation.

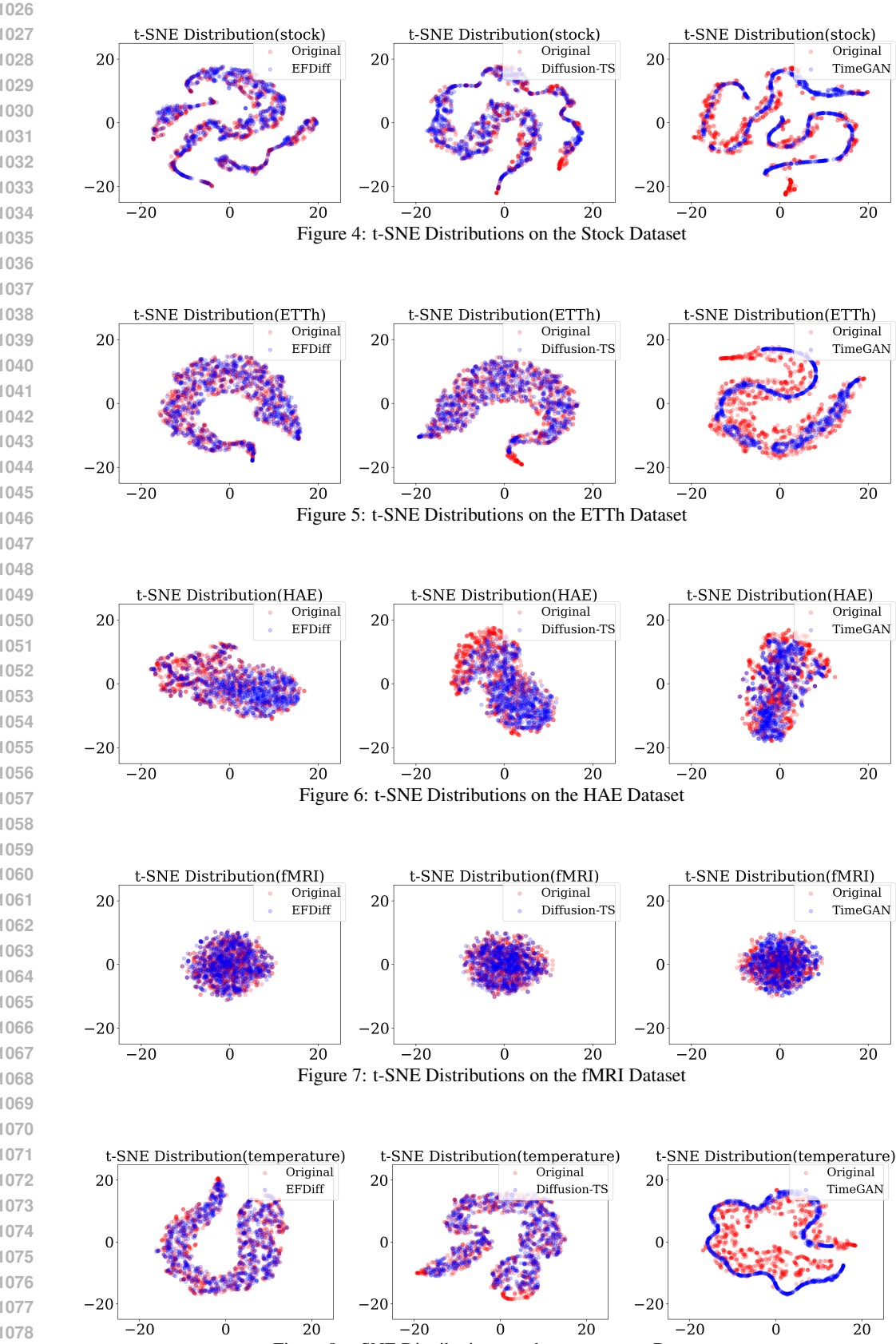

Figure 4: t-SNE Distributions on the Stock Dataset

Figure 5: t-SNE Distributions on the ETTh Dataset

Figure 6: t-SNE Distributions on the HAE Dataset

Figure 7: t-SNE Distributions on the fMRI Dataset

Figure 8: t-SNE Distributions on the temperature Dataset

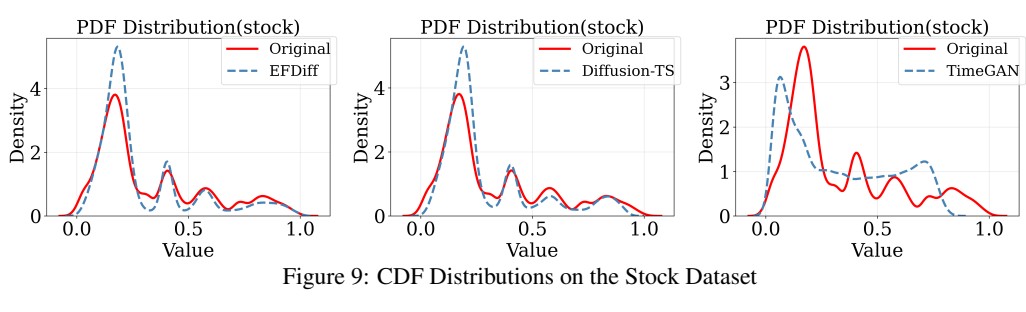

Figure 9: CDF Distributions on the Stock Dataset

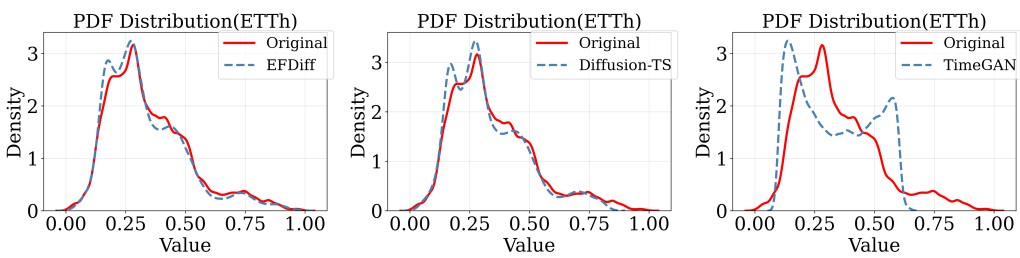

Figure 10: CDF Distributions on the ETTh Dataset

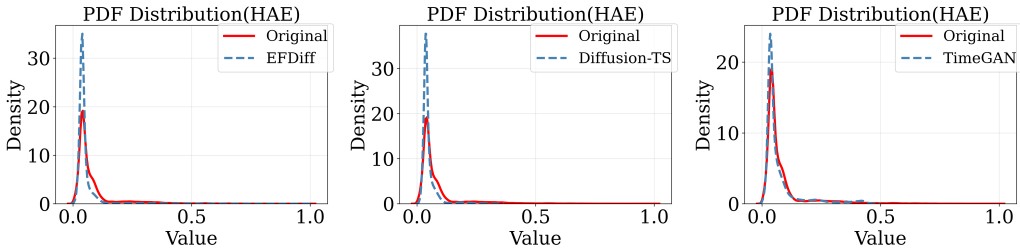

Figure 11: CDF Distributions on the HAE Dataset

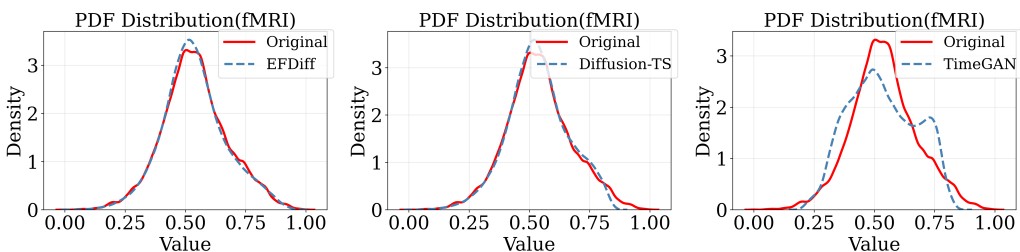

Figure 12: CDF Distributions on the fMRI Dataset

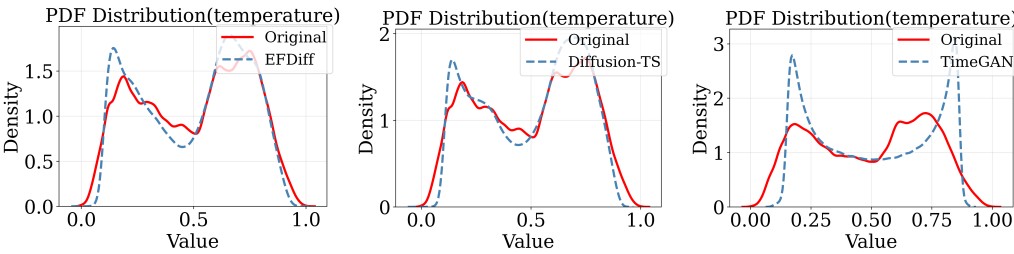

Figure 13: CDF Distributions on the Temperature Dataset

