# OpenReview forum: "EFDiff: Frequency-informed Diffusion for Extreme-value Time Series Generation"
_ICLR.cc/2026/Conference — Submitted to ICLR 2026_

### Official Review · Reviewer_YMnf · 2025-10-27

**Soundness:** 2
**Presentation:** 3
**Contribution:** 2
**Rating:** 4
**Confidence:** 3

**Summary:**

This paper proposes EFDiff a frequency-informed diffusion model for time series generation. In order to tackle the extreme value aware time-series generation, EFDiff has EFX and EFGEN module to enhance the generation quality.

**Strengths:**

1. This paper addresses the extreme value awareness in the unconditional time series generation problem, which is usually overlooked.
2. This paper carefully discussed what an extreme value means in the encoded latent space.
3. A good amount of details is provided in the methodology section.

**Weaknesses:**

1. While the model aims to optimize the extreme value awareness, there is a lack of suitable evaluation metrics for such scenarios. It is not clear how EFDiff works in such a case.
2. The dataset selected seems to be low-dimensional; it is unclear how they may perform in a high-dimensional dataset like energy (which is commonly used for unconditional generation).
3. The proposed extreme components have a couple of modules. A comprehensive ablation study could provide additional evidence for the necessity of each proposed module.

**Questions:**

1. The evaluation is mainly conducted over the general distribution. Another commonly used evaluation metric is discriminative accuracy (TimeGAN, TimeVAE, Diffusion-TS, etc.). Will the awareness of extreme values potentially harm the discriminative accuracy?
2. Is there a particular reason why PODT is calculated over Xenc? It seems that defining extreme value in the raw space is more natural than in the latent space.
3. How costly is this additional module in terms of time?

---

### Official Review · Reviewer_tY28 · 2025-10-30

**Soundness:** 1
**Presentation:** 2
**Contribution:** 1
**Rating:** 2
**Confidence:** 4

**Summary:**

This paper introduces EFDiff (Extreme-Frequency Diffusion), a frequency-informed diffusion model specifically designed for extreme-value time-series generation. The core novelty lies in a dedicated Extreme Component comprising two modules: (1) Extreme-Frequency Extraction (EFX), which heuristically constructs a global dictionary of extreme-contributing frequencies via an event-driven, multi-metric analysis; and (2) Extreme-Frequency Generation Enhancement (EFGEN), a Transformer-based soft frequency selector that adaptively models these extreme patterns during the denoising process. Extensive experiments on multiple public datasets demonstrate the proposed approach's empirical superiority in faithfully capturing extreme-value structures.

**Strengths:**

- The authors tackle the important yet underexplored problem of robustly modeling extreme-value distributions in time series, a domain with significant real-world implications.

- The framework is logically structured around its two main components - EFX for dictionary construction and EFGEN for adaptive frequency selection - providing a clear mechanism for frequency-informed generation.

- The approach demonstrates consistent empirical improvements across multiple public datasets, particularly on tail-sensitive metrics (e.g., KL and JS divergence), validating its specific strength in capturing rare, high-impact events.

**Weaknesses:**

- Although the authors present a data-driven analysis linking extreme events to the phase alignment of high-frequency components, the evidence is primarily observational. The core formulation lacks theoretical derivation or rigorous statistical validation, making the crucial connection between specific frequency-phase structures and extreme-value formation insufficiently justified.

- The design of the EFX module relies on a set of heuristic metrics (PLV, Cw, Lw) whose theoretical necessity and connection to extreme-value formation remain unclear. While a limited ablation study is provided in the appendix, it offers minimal insight into the relative contribution or necessity of each metric. Consequently, the formulation appears empirically motivated rather than theoretically grounded.

- The reported performance of the baseline FIDE (an existing extreme-aware diffusion framework) is substantially lower than both its original paper and even the performance of simpler models like Diffusion-TS. This unexpected inversion strongly suggests possible inconsistencies in dataset selection or experimental configuration. As a result, the fairness and reliability of the comparative evaluation are seriously questionable, making the claimed performance gains difficult to interpret objectively.

- The overall novelty is arguably incremental compared to existing frequency-based diffusion frameworks (e.g., FIDE and Diffusion-TS). The "Extreme Component" largely represents an extension rather than a fundamentally new generative paradigm. Furthermore, the paper’s exposition is often verbose and unclear, using overly technical terminology with limited visual or intuitive explanation of the proposed mechanism.

**Questions:**

Please refer to the Weaknesses part.

---

### Official Review · Reviewer_Wbut · 2025-10-30

**Soundness:** 2
**Presentation:** 2
**Contribution:** 2
**Rating:** 4
**Confidence:** 4

**Summary:**

This paper introduces EFDiff, a frequency-informed diffusion model tailored for realistic and extreme-value time series generation. The central innovation is the explicit extraction and modeling of extreme-contributing frequencies using a two-stage approach. Namely,
1. Extreme-Frequency Extraction (EFX), which creates a global dictionary of frequency combinations aligned with extreme events based on event-driven local analysis and multiple custom metrics.
2. Extreme-Frequency Generation Enhancement (EFGEN), which leverages a Transformer-based selection network to modulate the generation process by prioritizing those frequencies during the diffusion denoising steps.

Comprehensive experiments are conducted across five diverse real-world datasets and six evaluation metrics, showing EFDiff achieves strong performance in overall and extreme-value distribution fidelity.

**Strengths:**

The paper addresses a genuinely important and under-explored challenge: generating time series that authentically capture rare/extreme events rather than merely “smooth” approximations, a critical issue in domains like climate science and finance. The decomposition into trend, seasonality, and explicitly modeled extreme components and precise mathematical formulation of each step, especially the definition and scoring of extreme-contributing frequencies. The use of a Transformer-based soft frequency selection (EFGEN) is thoughtfully justified for its flexibility in frequency combination selection, and offers potential for extensibility.

**Weaknesses:**

While the contributions above, some key recent works that deeply involve frequency-domain diffusion (especially those fully formulating time series generation directly in the frequency domain) are not cited or contrasted:

Crabbé, Jonathan, et al. "Time series diffusion in the frequency domain." Proceedings of the 41st International Conference on Machine Learning. 2024.

Chi, Guoxuan, et al. "RF-diffusion: Radio signal generation via time-frequency diffusion." Proceedings of the 30th Annual International Conference on Mobile Computing and Networking. 2024.

The technical distinction of EFDiff vs. prior frequency-domain diffusion models is somewhat muddied. For example, extreme-aware frequency generation, dictionary construction, and frequency-based selection are quite similar in recent works (e.g. TS-Diff, and the above mentioned RF-diffusion).

The boundary between "extreme" and simply "high-frequency noise" is not sharply addressed, and the dictionary building process could be more rigorously described.

The paper heavily relies on the idea of phase alignment, yet the precise definition of "nearly aligned phases" is not fully formalized. Additionally, while amplitudes and phases are extracted from DFT/IDFT in the extreme component, there is minimal discussion regarding how uncertainty, non-stationarity, or multivariate extensions affect this decomposition.

The comparison suite is lengthy, but experiments are missing recent and directly comparable frequency-domain diffusion models (Crabbé 2024, Chi 2024, Gao 2024)

Gao, Jiaxin, Qinglong Cao, and Yuntian Chen. "Auto-regressive moving diffusion models for time series forecasting." Proceedings of the AAAI Conference on Artificial Intelligence. Vol. 39. No. 16. 2025.

Generally, the paper positions itself primarily against time-domain and partially frequency-aware models, but omits closely related frequency-domain diffusion papers, even though the text explicitly frames its contributions in the frequency domain. This makes it difficult to see exactly what is unique beyond combining known decomposition ideas with a new frequency‑scoring triplet and a soft‑selection module. The novelty is moderate, primarily due to the combination of event-driven frequency dictionaries and soft selection within diffusion, as well as the specific triad of frequency-contribution scores. The distinctiveness versus contemporaneous frequency-domain diffusion work needs to be argued and evaluated more directly.

**Questions:**

Can the authors provide more detail on the construction and selection process for frequency combinations in EFX? How is the combinatorial explosion of possible combinations managed? Are there tradeoffs in computational cost or redundancy?

What are the computational and sensitivity implications of the joint scoring ($S_w$) with learned $\alpha$, $\beta$ superpositions?

When choosing K-top combinations, do stability or redundancy issues arise, and how were they resolved in practice?

What is the value or motivation behind the threshold $\varepsilon$? Is there an adaptive mechanism?

---

### Official Review · Reviewer_gcpa · 2025-11-03

**Soundness:** 3
**Presentation:** 2
**Contribution:** 2
**Rating:** 6
**Confidence:** 4

**Summary:**

EFDiff proposes a frequency-informed diffusion framework for extreme-value time-series generation. The authors argue that conventional models fail to preserve extreme events due to smoothing bias and lack of frequency-domain awareness. Their method decomposes time series into trend, seasonality, and an explicit “Extreme Component.” This component is built via two modules: Extreme-Frequency Extraction (EFX), which constructs a global dictionary of frequency combinations characterizing extreme events using a novel dynamic thresholding strategy (PODT) and three contribution metrics; and Extreme-Frequency Generation Enhancement (EFGEN), a Transformer-based soft selector that retrieves relevant frequencies during denoising. Experiments on five real-world datasets show improved fidelity in both overall and extreme-value distributions, particularly in KL divergence.

**Strengths:**

1. The paper provides a principled frequency-domain perspective on extreme-value generation, grounded in empirical observation that phase-aligned high-frequency components drive extremal behavior—an insight not well exploited in prior time-series generative models.

2. The proposed EFX module introduces a multi-metric, phase-aware scoring mechanism that combines PLV, amplitude contribution, and background contrast, offering a more nuanced view of frequency relevance than simple amplitude-based selection.

**Weaknesses:**

1. The theoretical foundation for why specific frequency combinations cause extremes is underdeveloped; the paper asserts phase alignment matters but offers no formal proof or rigorous signal-theoretic justification beyond illustrative examples.

2. The PODT thresholding method, while adaptive, lacks comparison to standard EVT baselines like GEV or GP fitting, making it unclear whether the claimed “superior extreme identification” is merely an artifact of an arbitrary threshold design.

3. EFGEN’s “soft frequency selection” is essentially a standard cross-attention mechanism over a precomputed dictionary, offering negligible architectural novelty—this is repackaged retrieval, not a breakthrough in diffusion conditioning.

4. The ablation study fails to isolate the impact of the frequency-domain formulation itself; no baseline is tested that applies EFX-style metrics in the time domain, so the claimed superiority of the frequency view remains unsubstantiated.

5. Several equations are ambiguously defined: in Eq. (17), the summation over $f^{text}∈F^{text}$ conflates scalar frequencies with vector-valued combinations, and the IDFT implementation is never reconciled with the continuous cosine superposition in Eq. (3). It's a serious notational inconsistency.

6. The paper claims 55.7% KL improvement on Stocks but omits statistical significance testing or confidence intervals across seeds, casting doubt on the robustness of the reported gains.

**Questions:**

Please refer to the Weaknesses section for critical concerns that must be addressed regarding theoretical grounding, evaluation protocol, methodological novelty, and mathematical consistency.

---

> ### Comment · Reviewer_gcpa · 2025-11-26
>
> Considering that the authors did not address any of my concerns throughout the rebuttal period, I have decided to lower my score.

---

### Meta-Review · Area_Chair_akNt · 2025-12-15

**Summary:**

This submission proposes EFDiff, a diffusion-based framework for extreme-value time series generation using a frequency-informed “Extreme Component.” The method decomposes prediction into trend + seasonality + extreme components (following Diffusion-TS for the first two) and focuses novelty on: (i) EFX, which constructs a global dictionary of “extreme-contributing” frequency combinations using a hybrid extreme detector PODT and three heuristic contribution scores (PLV/Cw/Lw); and (ii) EFGEN, a Transformer module that performs soft selection over the dictionary to inject selected frequencies during denoising.

While one reviewer initially scored the paper slightly above threshold, the overall reviewing outcome is low/negative (two borderline rejects and one reject, plus an explicit post-rebuttal score decrease), driven by the following recurring concerns:
	•	Unclear novelty and positioning relative to prior frequency-domain diffusion: Reviewers note missing or insufficiently contrasted related work (e.g., frequency-domain diffusion approaches), and characterize EFGEN as largely standard attention-based retrieval over a precomputed frequency dictionary rather than a clear architectural innovation. This affects the perceived contribution beyond combining decomposition + frequency scoring + diffusion conditioning.
	•	Weak theoretical / methodological grounding for the key premise: The paper’s core claim—extremes arise from superposition of high-frequency components under phase alignment—is presented as largely observational, with limited formal justification. In addition, EFX relies on multiple heuristics (PLV/Cw/Lw and PODT) whose necessity is not theoretically established.
	•	Evaluation shortcomings and concerns about fairness/robustness: Reviewers raise issues about (a) missing directly comparable frequency-domain diffusion baselines, (b) unclear robustness/statistical significance of reported gains, and (c) questionable baseline implementations—especially the unexpectedly poor performance of FIDE compared to prior reports, which undermines confidence in the comparative conclusions. The experimental protocol defines “extremes” via PODT within length-24 blocks and top-10% selection, which may confound EVT-related claims and makes it hard to compare to standard EVT baselines.
	•	Ambiguity / inconsistency in mathematical definitions: At least one reviewer flags notational inconsistencies in how frequency combinations are summed and reconstructed (e.g., Eq. (17) vs the earlier cosine superposition framing), which reduces confidence in correctness and clarity of the frequency reconstruction pathway.
	•	Limited scope of ablations/diagnostics: While an ablation removing PLV/Cw/Lw is included, reviewers argue it does not isolate whether the frequency-domain framing itself is essential (e.g., time-domain analogs or alternative extreme definitions are not explored), and key sensitivity questions (dictionary size, combination explosion, thresholding alternatives) remain insufficiently answered.

Given these outstanding issues, particularly novelty/positioning and evaluation reliability, the paper is not recommended for acceptance despite tackling an important problem.

**Reviewer Concerns:**

Concerns that appear addressed (partially) by rebuttal / revision

Based on the PDF and review updates, some concerns are partially mitigated:
	•	Some ablation evidence for EFX metrics: The paper reports ablations removing PLV/Cw/Lw and shows degraded tail metrics, suggesting these heuristics matter within the proposed pipeline.
	•	Broader evaluation across multiple datasets and metrics: The paper includes five datasets and both tail-focused distribution metrics (KL/JS/Wasserstein/CRPS) and downstream proxies (predictive score, context-FID) in the appendix tables.
	•	Documentation/reproducibility claim: The manuscript states code and data are provided in an anonymous repository.

Concerns still outstanding (core reasons to reject)
	•	Novelty vs prior frequency-domain diffusion remains unclear: The paper’s related work section discusses some diffusion and extreme-aware work, but reviewers specifically pointed out missing/insufficient engagement with closely related frequency-domain diffusion lines; this remains a key gap because EFDiff’s headline contribution is “frequency-informed.”
	•	Theoretical justification of the “phase alignment → extremes” mechanism is not established: The method formalizes the assumption (Eq. (3)) but does not provide rigorous support beyond qualitative/illustrative evidence; as a result, the proposed mechanism reads as plausible but not well-founded.
	•	Extreme definition and EVT baseline comparisons: Extremes are defined using the paper’s PODT heuristic (max of several within-block thresholds) and then “top 10% within blocks,” but there is no clear comparison to standard EVT procedures (GEV/GP fits) or sensitivity to these design choices—exactly the concern raised by reviewers.
	•	Evaluation reliability and fairness concerns persist: The unexpectedly weak FIDE baseline performance (relative to what reviewers expected) remains unexplained in the manuscript; without careful reproduction notes or direct comparison protocols, the claimed improvements are difficult to interpret confidently.
	•	Mathematical clarity issues: The reconstruction definition for the Extreme Component (Eq. (17)) and the handling of frequency combinations vs individual frequencies are easy to misread and were explicitly flagged as inconsistent by reviewers; the paper does not convincingly resolve these notation/definition issues.
	•	Ablations do not isolate the value of “frequency-domain perspective” itself: The provided ablations remove metrics, but do not test comparable time-domain scoring/conditioning alternatives or alternative extreme extraction choices, leaving the central positioning claim under-supported.

**Reviewer Scores:**

Given the above, here is the likely score evolution if reviewers fully participated in discussion:
	•	Reviewer gcpa: Already explicitly lowered their score post-rebuttal due to concerns not being addressed. Their initial 6 would likely drop to 4 or 5 depending on how strongly they weigh the unresolved novelty/math/evaluation issues; their comment suggests 4–5, leaning 4.
	•	Reviewer Wbut (Score 4): Likely stays at 4. Their main issues (missing key related work comparisons, unclear distinction vs frequency-domain diffusion, unclear extreme vs high-frequency noise, combinatorial dictionary questions) are not convincingly resolved.
	•	Reviewer tY28 (Score 2): Likely stays at 2. Their strongest objection is evaluation reliability/fairness (baseline inversion) and weak theoretical grounding; neither is clearly resolved.
	•	Reviewer YMnf (Score 4): Likely remains at 4 (or possibly 3) given continued concerns about evaluation tailored to “general distribution,” lack of extreme-specific evaluation clarity, low-dimensional datasets (feature dimension 1 in configs), and missing comprehensive ablations. The paper’s reported setup indeed uses univariate sequences in the main config table, reinforcing the scalability concern.

---

### Decision · Program_Chairs · 2026-01-26

Reject